# The integration of Gaussian noise by long-range amygdala inputs in frontal circuit promotes fear learning in mice

Mattia Aime[†], Elisabete Augusto[†], Vladimir Kouskoff, Tiago Campelo, Christelle Martin, Yann Humeau, Nicolas Chenouard, Frederic Gambino*

University of Bordeaux, CNRS, Interdisciplinary Institute for Neuroscience, IINS, Bordeaux, France

**Abstract** Survival depends on the ability of animals to select the appropriate behavior in response to threat and safety sensory cues. However, the synaptic and circuit mechanisms by which the brain learns to encode accurate predictors of threat and safety remain largely unexplored. Here, we show that frontal association cortex (FrA) pyramidal neurons of mice integrate auditory cues and basolateral amygdala (BLA) inputs non-linearly in a NMDAR-dependent manner. We found that the response of FrA pyramidal neurons was more pronounced to Gaussian noise than to pure frequency tones, and that the activation of BLA-to-FrA axons was the strongest in between conditioning pairings. Blocking BLA-to-FrA signaling specifically at the time of presentation of Gaussian noise (but not 8 kHz tone) between conditioning trials impaired the formation of auditory fear memories. Taken together, our data reveal a circuit mechanism that facilitates the formation of fear traces in the FrA, thus providing a new framework for probing discriminative learning and related disorders.

*For correspondence:
frederic.gambino@u-bordeaux.fr

[†]These authors contributed equally to this work

Competing interests: The authors declare that no competing interests exist.

## Introduction

Discriminative learning is an important survival strategy that depends on the repeated contingency and contiguity between sensory cues (conditioned stimuli, CS) and the events (e.g. danger, safety) that they must predict (unconditioned stimuli, US) (*Hall, 2002*). It has been classically studied by using differential fear-conditioning paradigms where two different auditory CS are positively (CS+) and negatively (CS-) paired in time with an aversive US (e.g. foot shock). This learning protocol is supposed to assign appropriate emotional valence to the two incoming CSs (*Hall, 2002*; *LeDoux, 2000*; *Likhtik and Paz, 2015*). However, while previous work has thoroughly investigated how CSs modulate fear responses after learning (*Dejean et al., 2016*; *Karalis et al., 2016*; *Likhtik et al., 2014*; *Rogan et al., 2005*; *Stujenske et al., 2014*), it remains unclear how the brain processes and encodes CS+ and CS- during conditioning.

The medial prefrontal cortex (mPFC) has appeared over the past decade as a critical region that shapes behaviors in response to both aversive and non-aversive environmental cues (*Likhtik and Paz, 2015*; *Likhtik et al., 2014*; *Stujenske et al., 2014*). These effects of the mPFC possibly rely on the specific interaction between its different subdivisions (i.e.prelimbic [PL] and infralimbic [IL] cortices) and the basolateral complex of the amygdala (BLA) (*Senn et al., 2014*; *Sierra-Mercado et al., 2011*; *Vidal-Gonzalez et al., 2006*). However, the mPFC does not receive direct sensory information neither from sensory cortical areas or from the thalamus (*Hoover and Vertes, 2007*), thereby supporting the idea that a higher order neuronal network above the mPFC might encode specific memories that are later selected preferentially during recall together with its downstream cortical (e.g. PL or IL mPFC) or subcortical structures (e.g. BLA). Specifically, the frontal association cortex (FrA), a region of the lateral part of the agranular cortex (*Paxinos and Franklin, 2001*; *Uylings et al., 2003*),

receives inputs from the BLA (*Lai et al., 2012*; *Mátyás et al., 2014*; *Nakayama et al., 2015*) and sensory cortices (*Hoover and Vertes, 2007*; *Zhang et al., 2016*). In addition, the FrA appears non-reciprocally connected to the PL/IL subdivisions of the mPFC (*Zhang et al., 2016*). Thus, the FrA may function as a relay station during learning from sensory cortical areas and the BLA to the mPFC. However, whether and how the FrA integrates the variety of sensory information required for discriminative learning is not understood.

The involvement of the FrA in auditory fear conditioning has constantly been reported. For example, the pharmacological inactivation of FrA neurons alters both the expression and extinction of learned fear (*Lai et al., 2012*; *Nakayama et al., 2015*; *Sacchetti et al., 2002*). Recently, fear conditioning and extinction have been shown to induce dendritic spine elimination and formation in the FrA, respectively (*Lai et al., 2012*). Importantly, this phenomenon occurs within the same dendritic branch, supporting the idea that a unique FrA circuit could form memory traces with distinct emotional values. Nonetheless, no previous evidence has demonstrated the contribution of the FrA in the encoding of incoming sensory cues during learning, and, if so, how such a process may be controlled by inputs from the BLA.

To address the possible role of the FrA in discriminative learning and the mechanisms behind it, we investigated auditory-evoked computations of layers II/III FrA pyramidal neurons, as well as the dynamics of long-range projections from the BLA during the acquisition of fear memory traces. By using two-photon (2P) calcium imaging in head-restrained mice, in vivo whole cell recordings and optogenetics, we found that FrA pyramidal neurons process auditory tones based on their spectral properties. Unlike pure frequency tones, Gaussian noise produced somatic and dendritic depolarizations in FrA pyramidal neurons. The photo-stimulation of BLA-to-FrA neurons resulted in the supra-linear integration of auditory tones. During conditioning, the activity of BLA-to-FrA axons was stronger between CS+/US pairings (that is, when CS- is presented) than during pairings. Inhibiting these axons during CS- impaired auditory fear learning but only when Gaussian noise is used as CS-. Taken together, our data suggest that FrA and BLA-to-FrA neurons promote learning by integrating non-conditioned Gaussian noise (i.e. not paired to the footshock), and provide additional evidence for a critical function of cortical areas when animals learn from complex tones (*Grosso et al., 2015*; *Letzkus et al., 2011*; *Ohl et al., 1999*). In conclusion, the study reveals a potent dendritic mechanism for encoding predictors in the FrA, and thus extends the cortical framework for probing discriminative learning and related disorders.

## Results

### Auditory tones recruit NMDARs conductances in FrA L2/3 pyramidal neurons

We performed somatic whole-cell recordings in anesthetized naive animals and characterized the activation of FrA L2 pyramidal cells to different sounds by presenting white Gaussian noise (WGN), pure frequency tones (4 and 8 kHz), as well as linear chirps of these frequencies (*Figure 1*; *Figure 1—figure supplement 1*). WGN, which is a true random signal with equal intensity at all frequencies (*Figure 1—figure supplement 1*), and pure frequency tones have been widely used in previous fear-conditioning studies (*Dejean et al., 2016*; *Grosso et al., 2015*; *Karalis et al., 2016*; *LeDoux, 2000*; *Likhtik et al., 2014*; *Park et al., 2016*; *Senn et al., 2014*; *Stujenske et al., 2014*).

For each recorded cell, membrane potential (Vm) was monitored prior to, during, and after the random presentation of both tones, each consisting of 27 pips (50 ms, 0.9 Hz for 30 s) (*Figure 1B–F*). Membrane potential spontaneously fluctuated between up and down states (*Figure 1B*). Therefore, to detect changes in Vm specifically induced by the auditory stimulation, we computed the cumulative depolarization over time (cVm), from which was then subtracted the linear regression calculated during the baseline period prior to auditory stimulation (cVm change) (*Figure 1C,D*; *Figure 1—figure supplement 2A,B*). This allowed us to minimize the variability related to spontaneous activity and thus compare evoked depolarizations under different conditions.

WGN evoked a subthreshold depolarization in naive animals (27.6 ± 4 mV, n = 22 cells) that lasted for at least 30 s after the end of the stimulation (32.5 ± 5 mV, n = 22 cells) (*Figure 1C,D*). These changes in Vm, which were always higher than those evoked by the linear chirps or the pure tones (*Figure 1—figure supplement 1*), were abolished by the application of the NMDAR antagonist D(-

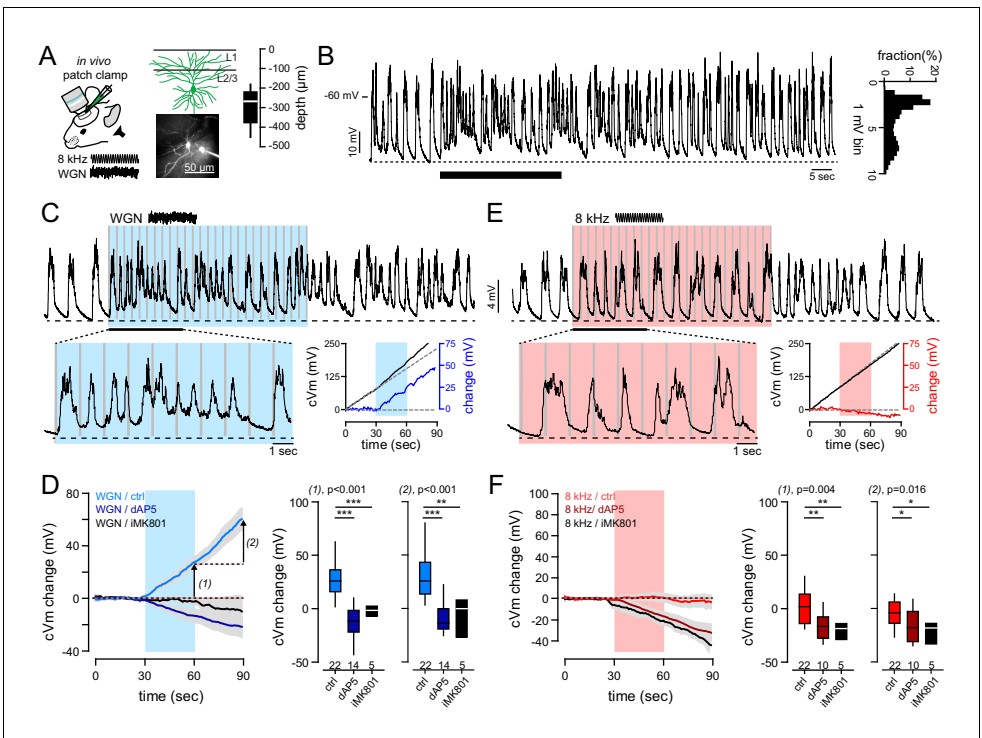

**Figure 1.** Gaussian auditory stimulation depolarizes FrA pyramidal neurons. (**A**) Membrane potential (Vm) was recorded from L2/3 FrA pyramidal under urethane anesthesia with 2P visual guidance. Depth of recorded cells is indicated. (**B**) Example of recorded FrA neuron showing typical spontaneous slow wave fluctuations. Black bar indicates timing of Gaussian stimulation. *Right*, membrane potential probability histogram. (**C**) Example traces of postsynaptic membrane potential recorded from an individual FrA L2/3 pyramidal neuron upon white Gaussian noise (*WGN*) auditory stimulation (gray bars: 27 pips, 50 ms in duration, 0.9 Hz, 30 s). *Bottom right panels*, auditory-evoked cVm changes were calculated by subtracting the cumulative Vm (cVm, solid black line) by its linear regression during the baseline period (solid black line). (**D**) *Left*, averaged cVm change (±sem) evoked by *WGN*, with or without the blockade of NMDARs (dAP5 or iMK801). Auditory stimulation is depicted by the blue bar; arrows, analysis time points 1 (end of stimulation) and 2 (30 s later). *Right*, effect of dAP5 and iMK801 on cVm change at time points 1 (left) and 2 (right). Boxplots represent median and interquartile range. (**E, F**) Same representation as (**C, D**) but for pure 8 kHz-evoked cVm change. The effect of both stimuli was tested on the same cell. \*\*\*, p<0.001; \*\*, p<0.01; \*, p<0.05.

The online version of this article includes the following source data and figure supplement(s) for figure 1:

**Source data 1.** Sound-evoked depolarization.

**Figure supplement 1.** Spectral properties of the auditory tones used in our study, and their effect on membrane potential of FrA L2/3 pyramidal neurons.

**Figure supplement 1—source data 1.** Spectral and sound pressure properties.

**Figure supplement 2.** dAP5 application affects spontaneous and auditory-evoked FrA membrane potential changes.

**Figure supplement 2—source data 1.** Effect of dAP5 application.

)−2-amino-5-phosphonovaleric acid (dAP5) or the presence of the NMDAR open-channel blocker MK-801 in the intracellular solution (end of the stimulation; control: 27.6 ± 4 mV, n = 22 cells; +dAP5: −13.4 ± 5 mV, n = 14 cells; +iMK801: −2.2 ± 2 mV, n = 5 cells; p<0.001, *anova*) (*Figure 1D*; *Figure 1—figure supplement 2C–F*), suggesting that WGN generates NMDAR-mediated depolarization.

In contrast, pure auditory tones did not seem to affect membrane potential either during (*8 kHz*: 1.2 ± 4 mV; *WGN*: 27.6 ± 4 mV; n = 22 cells, p<0.001, paired t-test) or after auditory stimulation (*8 kHz*: −3.8 ± 3 mV; *WGN*: 32.5 ± 5 mV; n = 22 cells, p<0.001, *paired t-test*). Nonetheless, we found that 8 kHz tones affected membrane potential of FrA neurons in the presence of dAP5 or iMK801 (end of stimulation; control: 1.16 ± 3.9 mV, n = 22 cells; +dAP5: −16.7 ± 4 mV, n = 10 cells;

+iMK801: −21 ± 3.5 mV, n = 5 cells; p=0.004, *anova*) (*Figure 1C–F*). This suggests that pure tones might also activate FrA neurons to some extent. However, on average, the NMDAR-mediated component of the evoked cVm change was much larger in response to WGN, indicating that it recruited more NMDAR conductances than pure auditory tones (*WGN*: 33.1 ± 4.2 mV; *8 kHz*: 9.7 ± 3.4 mV; n = 5 cells; p=0.017, paired *t-test*) (*Figure 1—figure supplement 2G–J*). Altogether, our data indicate that, during anesthesia, WGN activates enough NMDAR-mediated synaptic inputs to produce sustained depolarization of the cell body.

## Auditory tones generate local calcium events in distal FrA dendrites

NMDARs conductances confer unique computational capabilities to pyramidal neurons by operating supra-linear signaling in dendrites (*Antic et al., 2010*; *Major et al., 2013*). These NMDAR-mediated events are highly localized to a small dendritic segment but can spread toward the soma to produce plateau potentials (*Gambino et al., 2014*; *Palmer et al., 2014*). Thus, we investigated whether auditory tones generate local dendritic events. We infected mice with an AAV9-Syn-flex-GCaMP6s together with a 1:10,000 dilution of AAV1-hSyn-cre (*Figure 2A*) in order to obtain a sparse labeling with few neurons expressing GCaMP6s. The activity of non-overlapping distal dendritic branches was imaged in superficial layer one through an implanted cranial window in anesthetized (139 dendrites from five mice; *Figure 2B–D*) and awake mice (104 dendrites from eight mice; *Figure 2B,E,F*). We isolated calcium transients and segregated them based on their spatial spread along individual dendrites (*Figure 2C*; *Figure 2—figure supplement 1A–C*).

We detected calcium transients in apical dendritic tufts of mice that were anesthetized with isoflurane (1.5%). These calcium events occurred both spontaneously (i.e. during baseline prior to stimulation) and upon auditory stimulation with similar amplitude (*Figure 2C,D*). The presentation of WGN evoked more events (*WGN*: 409; *8 kHz*: 139; *2 kHz*: 102; p<0.001, *McNemar's $\chi^2$ test*), with a significantly higher number of local dendritic events per dendrite as compared to pure frequency auditory tones (*WGN*: 0.11 ± 0.01 Hz; *8 kHz*: 0.04 ± 0.01 Hz; *2 kHz*: 0.04 ± 0.007 Hz; n = 9 mice; p<0.001, *anova*) (*Figure 2C,D*). Eight kHz and 2 kHz tones did not evoke calcium transients more frequently than baseline (*Figure 2D*). Although it remains possible that pure tones activate basal dendrites, our data demonstrate that, during anesthesia, apical dendrites located in L1 are specifically activated by WGN.

During anesthesia, auditory-evoked calcium events were mostly local, with a full width at half maximum (fwhm: 13.5 ± [s.d.] 13 µm, n = 650 events) that fell into the spatial range of NMDAR-mediated spikes (*Gambino et al., 2014*; *Palmer et al., 2014*; *Figure 2G*). Global calcium transients with a longer spatial extent (fwhm ≥50 µm; 81 ± [s.d.] 35 µm, n = 313 events) were additionally observed in awake mice (*Figure 2E,H*). These events, which could reflect backpropagating somatic action potentials, occurred independently of the nature of the auditory tone (*Figure 2—figure supplement 1*). In contrast, most of the local calcium transients were concurrent with *WGN* (*WGN*: 339; *8 kHz*: 218; *2 kHz*: 253; p<0.001, *McNemar's $\chi^2$ test*). Again, and as expected, WGN evoked significantly more local events per dendrite than pure tones (*WGN*: 0.11 ± 0.01 Hz; *8 kHz*: 0.08 ± 0.005 Hz; *2 kHz*: 0.07 ± 0.007 Hz; n = 5 mice; p<0.001, *anova*) (*Figure 2F*), thus mirroring what we observed during anesthesia (*Figure 2I*). Nevertheless, *8 kHz* and *2 kHz* tones generated more local events as compared to baseline (*Figure 2F*), suggesting that anesthesia might reduce dendritic signaling of pure frequency auditory inputs (0.04 ± 0.009 Hz *vs.* 0.08 ± 0.05 Hz; p=0.037) (*Figure 2I*). However, the specific effect of *WGN* on FrA neurons was not attributable to the intensity of the tones (*Figure 1—figure supplement 1E*), or to noise-induced pain since exploratory motor behaviors were not affected by *WGN* or pure tones (*Figure 2—figure supplement 2*).

Taken together, our results show that FrA pyramidal neurons process auditory tones differently according to their spectral properties in both anesthetized and awake mice. This occurs at the subthreshold level with Gaussian auditory tones being more efficient in producing somatic depolarizations and local dendritic events within the same tuft dendritic branch than pure frequency tones (*Figure 1*; *Figure 2*) or simple frequency-modulated tones (*Figure 1—figure supplement 1*).

## Non-conditioned Gaussian noise promotes fear memories in FrA circuit

WGN and pure frequency tones produce distinct forms of synaptic plasticity during auditory fear conditioning (*Park et al., 2016*). This raises the possibility that they might play a specific role during

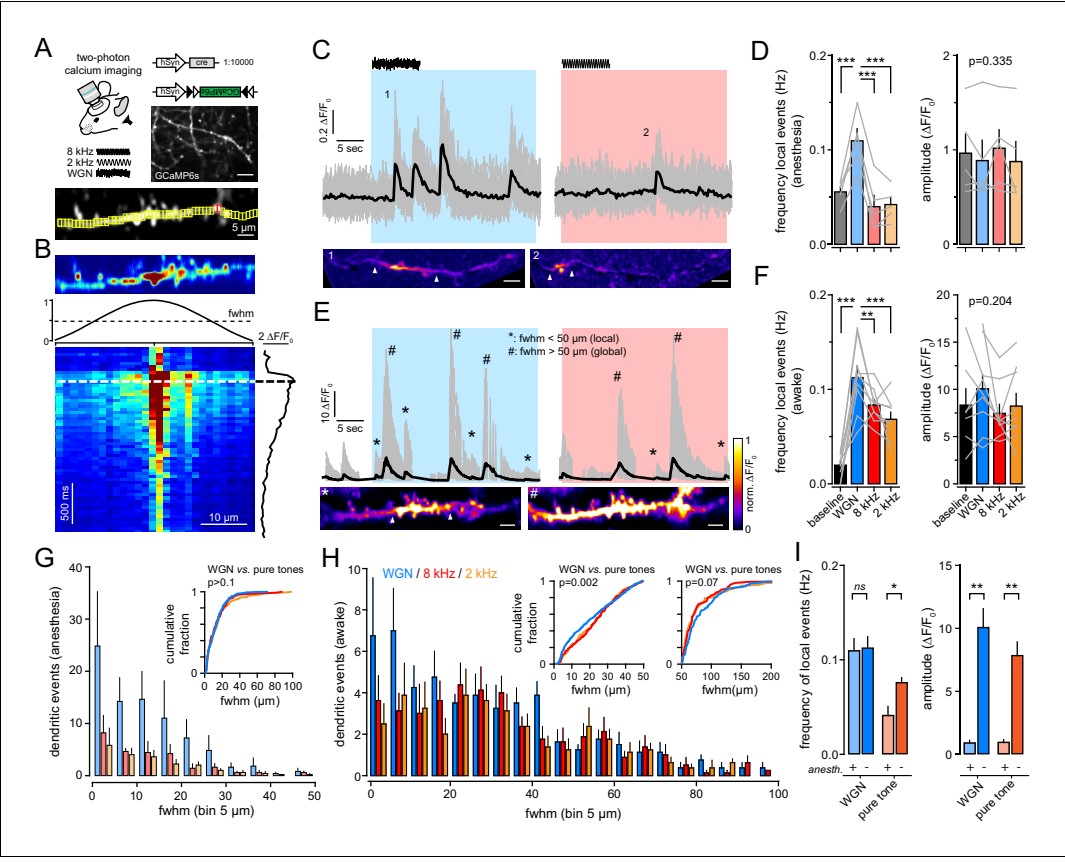

**Figure 2.** Gaussian auditory stimulation evokes more local dendritic events than pure frequency tones in anesthesia and awake. (A) *Top*, experimental strategy resulting in the sparse expression of GCaMP6s. Scale bar, 10 μm. Ca²⁺ events in individual dendrites were imaged in L1 upon auditory stimulation in awake or anesthetized mice. *Bottom*, example of GCaMP6s fluorescence standard deviation image with ROIs in yellow. Red ROIs overlapped with another dendrite and were excluded from the analysis. (B) The spread of Ca²⁺ events was quantified by calculating the full-width at half-max (fwhm, dashed line) of the normalized Gaussian fit at the time when the averaged $\Delta F/F_0$ was maximal. (C) *Top*, example of multiple Ca²⁺ transients ($\Delta F/F_0$) recorded in a single dendrite in anesthetized mouse, during baseline and upon WGN (blue) and 8 kHz (red). Gray lines, $\Delta F/F_0$ measured from small ROIs (all ROIs superimposed). Black line, mean $\Delta F/F_0$ averaged over all ROIs, respectively. *Bottom*, examples of local dendritic Ca²⁺ events upon WGN (1) and 8 kHz (2). Scale bar, 5 μm. (D) The frequency (*left*) and amplitude (*right*) of all Ca²⁺ events observed during baseline and upon auditory stimulations was averaged over five anesthetized mice. Gray lines represent individual mice. (E) *Top*, same representation as (C) but in awake mouse. *Bottom*, example of local (\*) and global (#) dendritic Ca²⁺ events. Scale bar, 5 μm. (F) Same presentation as in (D) but for local events from nine awake mice. Gray lines represent individual mice. (G, H) Distribution histogram of dendritic events fwhm upon WGN (blue), 8 kHz (red) and 2 kHz (orange) tone presentation, from anesthetized (G) and awake (H) mice. *Inset*, cumulative probability histograms of auditory-evoked events fwhm. (I) Dendritic events evoked by WGN and pure tones (8 kHz and 2 kHz tones pooled together) in anesthetized (+) and awake (-) mice. \*, p<0.05; \*\*, p<0.01; \*\*\*, p<0.001.

The online version of this article includes the following source data and figure supplement(s) for figure 2:

**Source data 1.** Frequency of dendritic calcium events.

**Figure supplement 1.** Auditory-evoked calcium events in dendritic tufts are similar between brain states.

**Figure supplement 1—source data 1.** Number of dendritic calcium events.

**Figure supplement 2.** Auditory tones do not affect locomotor activity.

**Figure supplement 2—source data 1.** Locomotor activity.

learning (**Grosso et al., 2015**; **Park et al., 2016**). To test this hypothesis, we injected mice bilaterally with an AAV expressing the light-activated proton pump *archaerhodopsin* (AAV9.CaMKII.ArchT. GFP, n = 40 mice; or AAV9.CamKII.eGFP for controls, n = 35 mice) into the FrA. Although both L5 and L2/3 cells were infected (**Figure 3A**), we implanted optical fibers in the superficial L1 to inhibit the activity of FrA L2/3 pyramidal somas (as well as L5 dendrites) during fear learning (**Figure 3B**). Auditory fear conditioning (FC) was induced by using a classical discriminative protocol, during which five auditory stimuli (each consisting of 27 *WGN* or *8 kHz* pips, 50 ms, 0.9 Hz for 30 s) were positively (CS+) or negatively (CS-) paired with the delivery of a mild electrical shock (0.6 mA) to the paws in a pseudorandom order (**Figure 3—figure supplement 1**). The auditory tones (8 kHz and WGN) used for CS+ and CS- during conditioning were counterbalanced across mice (protocol 1, CS+/CS-: *8 kHz/WGN,* respectively; protocol 2, CS+/CS-: *WGN/8 kHz,* respectively) (**Figure 3—figure supplement 1**) and learning was tested 24 hr later during recall by measuring cue-induced freezing in a novel context (**Figure 3B**). Mice were classified as learners (learning+) when the learning index was higher than 20% during recall (**Figure 3—figure supplement 1**).

First, we suppressed the activity of FrA L2/3 neurons by delivering light each time a CS+ was presented during conditioning (**Figure 3C,D**). Surprisingly, neither the fraction of mice that learned the cue-shock association (protocol 1, GFP: 75%, n = 8 mice, ArchT: 87%, n = 8 mice, p=0.522; protocol 2, GFP: 85%, n = 7 mice, ArchT: 70%, n = 7 mice, p=0.248, *Pearson's $\chi^2$ test*), nor the percentage of freezing induced by CS+ during recall (protocol 1, GFP: 51.2 ± 8%, n = 8 mice; ArchT: 46.2 ± 8%, n = 8 mice; p=0.7; protocol 2, GFP: 45.3 ± 4%, n = 7 mice; ArchT: 33.3 ± 5%, n = 7 mice; p=0.105) were affected by the inhibition of FrA neurons (**Figure 3D–F**). In contrast, ArchT stimulation during CS- presentation (**Figure 3G,H**) significantly decreased the fraction of mice that were conditioned. However, this occurred only when *WGN* was used as CS- (protocol 1, GFP: 75%, n = 12 mice, ArchT: 30%, n = 13 mice, p=0.02; protocol 2, GFP: 85%, n = 7 mice, ArchT: 91%, n = 11 mice, p=0.7, *Pearson's $\chi^2$ test*) (**Figure 3I**). In those mice, the magnitude of CS+ evoked freezing responses was significantly lower during recall than in control mice (protocol 1, GFP: 51 ± 7%, n = 12 mice; ArchT: 24.5 ± 5%, n = 13 mice; p=0.005) (**Figure 3J**).

These data indicate that when *WGN* is paired negatively to footshock during conditioning, it promotes fear memory traces in the FrA circuit, thus confirming the specific nature of complex auditory cues during learning (**Grosso et al., 2015**; **Park et al., 2016**). To further clarify the synaptic mechanisms involved, we measured auditory-evoked dendritic and somatic responses following conditioning protocol 1. As compared to naive mice, *WGN* (CS-) generated less local dendritic activation during recall (protocol 1; *naive*: 0.11 ± 0.01 Hz, n = 9 mice; *FC*: 0.06 ± 0.009 Hz, n = 5 mice; p=0.003) (**Figure 3—figure supplement 2**). In contrast, local dendritic events evoked by a conditioned cue (8 kHz) or cues that were not presented during conditioning (2 kHz) were not altered, indicating that fear learning specifically affected WGN-mediated dendritic signaling. In addition, WGN-induced somatic plateau potentials were reduced in conditioned animals in a learning-dependent manner, suggesting that NMDAR-dependent plasticity mechanisms in FrA neurons were engaged during learning (**Lai et al., 2012**; **Rioult-Pedotti et al., 2000**; **Figure 3—figure supplement 3**). Interestingly, the somatic and dendritic responses to both 8 kHz and WGN tones were similar during recall while mice performed better at discriminating these stimuli. This suggest that, while dendritic signaling appears critical during learning to recruit pyramidal neurons into memory traces, other circuit and cellular mechanisms are at play to discriminate sensory cues after learning. Whether the effect of learning on FrA pyramidal neurons is cancelled following fear extinction remains unknown and would be interesting to explore.

## BLA-to-FrA axons are recruited during fear conditioning

The above-mentioned results indicate that the FrA is required for fear learning and guides behaviors by integrating non-conditioned complex auditory cues during conditioning. Given the well-established role of the BLA and its cortical projection during the acquisition and expression of auditory cue fear learning (**LeDoux, 2000**; **Likhtik and Paz, 2015**; **Likhtik et al., 2014**; **Nakayama et al., 2015**; **Senn et al., 2014**; **Stujenske et al., 2014**), we next investigated the information transmitted from the BLA to the FrA (**Lai et al., 2012**; **Mátyás et al., 2014**; **Nakayama et al., 2015**) during conditioning. We injected a virus expressing the genetically encoded calcium indicator GCaMP6f into the right BLA and imaged axonal Ca²⁺ responses in the superficial L1 of the right FrA of awake head-restrained mice during fear conditioning (**Figure 4A,B**). Results confirmed that BLA neurons

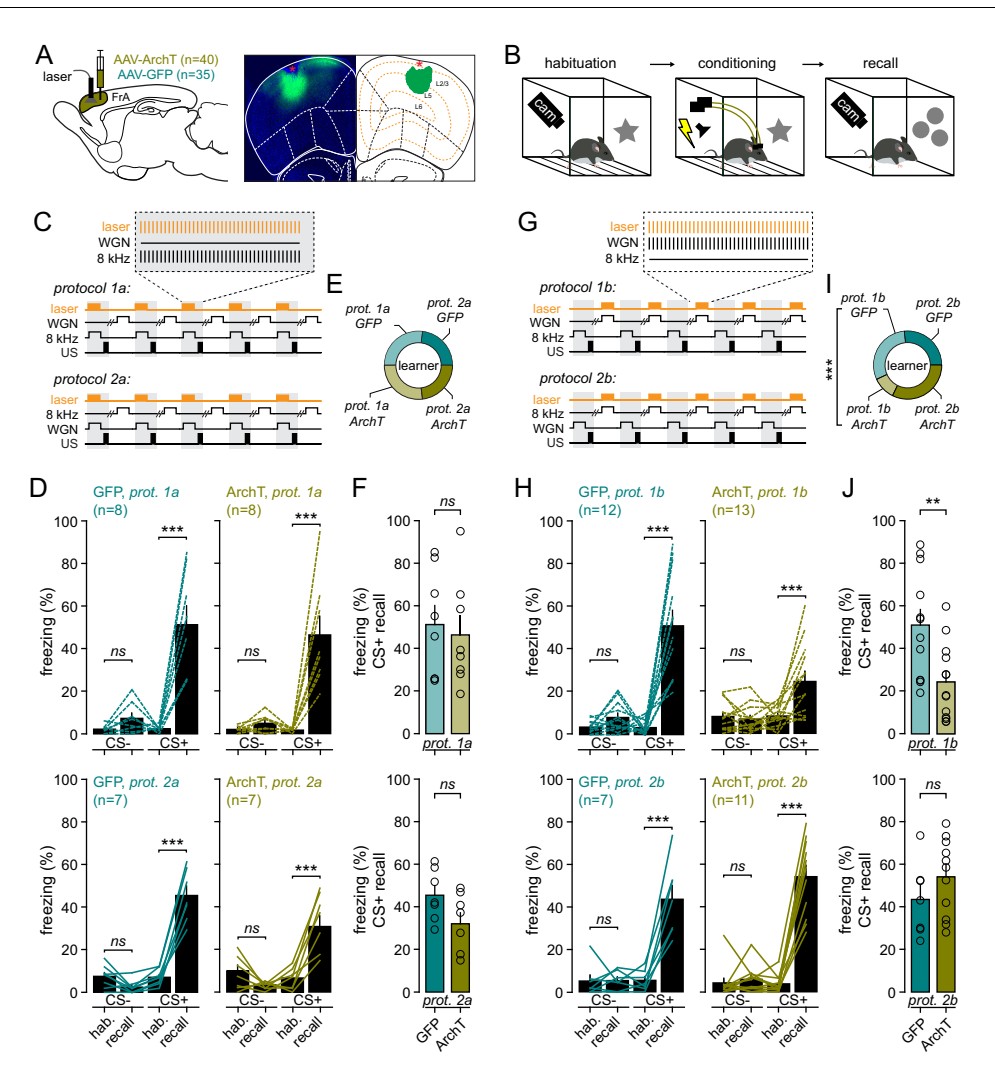

**Figure 3.** The FrA is engaged during fear learning when WGN is used as a CS-. (**A**) *Left*, experimental strategy. Mice were infected with AAV9.CaMKII.ArchT (n = 39) or AAV9.CaMKII.eGFP for controls (n = 34) in the FrA, and implanted bilaterally with optical fibers in the FrA. *Right*, expression profile of ArchT (in green) in the FrA (bregma: +2.58). DAPI staining in blue. *E*ntropy thresholding was used to determine the expression of ArchT across layers. (**B**) Timeline of auditory fear-conditioning behavioral protocol, with habituation and fear conditioning done in one context and fear learning quantified 24 hr later in a new context. (**C**) Experimental fear-conditioning protocols. FrA neurons expressing ArchT (or GFP) were photo-stimulated during the presentation of CS+ (8 kHz for protocol 1a or WGN for protocol 1b). US, unconditional stimulus (footshock). (**D**) Effect of light during learning on freezing behaviors during recall as compared to habituation (hab.) in GFP-expressing mice (left; n = 15) and ArchT-expressing mice (right, n = 15). Turquoise and khaki lines represent individual mice. (**E**) Fraction of mice that learned the cue-shocked association for each protocol. (**F**) Freezing behaviors quantified during CS+ in GFP- and ArchT-expressing mice. Circles, individual mice. (**G–J**) Same representation as (**C–F**) but for experiments with photo-stimulations delivered during CS-.

The online version of this article includes the following source data and figure supplement(s) for figure 3:

**Source data 1.** Freezing responses.
**Figure supplement 1.** Effect of FrA opto-inhibition during conditioning on freezing responses and learning during recall.
**Figure supplement 2.** Fear conditioning (protocol 1) specifically decreases WGN-induced local dendritic events.
**Figure supplement 2—source data 1.** Dendritic calcium events after fear learning.
**Figure supplement 3.** Fear conditioning (protocol 1) occludes WGN-evoked somatic plateau potentials.
**Figure supplement 3—source data 1.** Sound-evoked depolarization after fear learning.

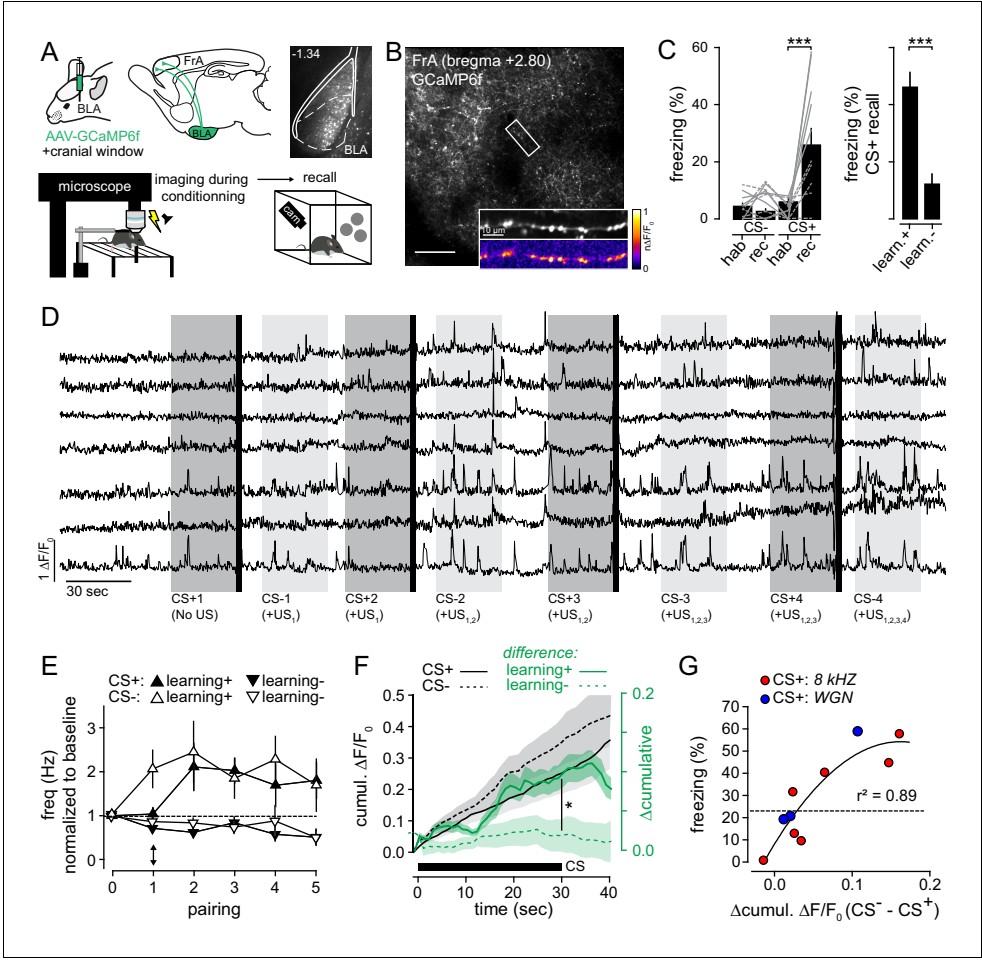

**Figure 4.** The activation of BLA-to-FrA axons between conditioning trials predicts the level of auditory fear learning. (A) *Top left*, experimental protocol. BLA-to-FrA axons were imaged in the superficial layer of the FrA. *Top Right*, expression profiles of GCaMP6f in the BLA. *Bottom*, GCaMP6f-expressing mice were fear conditioned under the 2P microscope (conditioning protocols 1 and 2 pooled together), and fear learning was quantified 24 hr later in a new context. (B) GCaMP6f-expressing axons were imaged in the FrA of awake mice (scale bar, 200 μm). *Inset*, example of $Ca^{2+}$ transients in individual BLA-to-FrA axon. (C) *Left*, conditioning under the microscope-induced robust fear behaviors: in contrast to CS-, CS+ increased freezing responses during recall as compared to habituation (hab.). Gray solid lines, % of freezing of mice with learning index >20% (learning+, n = 5); gray dashed lines, % of freezing of mice with learning index <20% (learning-, n = 5). *Right*, freezing responses induced by CS+ during recall for learners (learn. +) and non-learners (learn. -) mice. (D) Examples of $Ca^{2+}$ transients ($\Delta F/F_0$) from individual boutons recorded from one mouse upon consecutive CS+/US pairings. Dark gray bars, CS+; light grey bars, CS-; black bars, footshock (US). (E) Frequency of axonal $Ca^{2+}$ transients recorded during successive CS. For the first CS+, the activity of boutons was monitored in absence of footshock. Black arrow points to the difference between the first CS+ (before the first footshock) and the first CS- (after the first footshock). 0 corresponds to the baseline period before conditioning. (F) Cumulative $\Delta F/F_0$ averaged (±sem) across all CS+ (solid black lines, n = 10) and CS-(dotted black lines, n = 10). Green lines, difference between CS+ and CS- related axonal activity ($\Delta$cumulative: $\Delta F/F_{0\ CS-}$ - $\Delta F/F_{0\ cs+}$) in mice that learned (learning+, solid green line, n = 5) or not (learning-, dotted green line, n = 5) the association. (G) Relation between the $\Delta$cumulative during conditioning and the % of freezing during recall for protocol 1 (blue, CS+: 8 kHz) and protocol 2 (red, CS+: WGN). Circles, mice. *, p<0.05; ***, p<0.001.

The online version of this article includes the following source data and figure supplement(s) for figure 4:

**Source data 1.** Frequency of calcium events in axons.

**Figure supplement 1.** BLA-to-FrA axons do not encode the nature nor the valence of auditory tones.

project to the superficial layer 1 of the ipsilateral FrA (<150 µm from the pia) (*Lai et al., 2012*; *Mátyás et al., 2014*; *Nakayama et al., 2015*; *Figure 4B*; *Figure 4—figure supplement 1*). We conditioned awake mice (n = 10 mice) under the two-photon microscope by using the same counterbalanced protocols as described previously (protocol 1, n = 7 mice; protocol 2, n = 3 mice) (*Figure 3—figure supplement 1*). GCaMP6f calcium transients ($\Delta F/F_0$) provided a direct measure of the activation of BLA neurons projecting to the FrA (*Figure 4B*). Again, learning was tested 24 hr later and quantified by the percentage of freezing (*Figure 4C*). We then compared the activity of individual boutons between mice that learned (*learning+*, n = 5 mice) and those that failed to learn (*learning-*, n = 5 mice).

As a first metric to quantify the activity of BLA-to-FrA axons, we measured the number of calcium transients in individual boutons observed during baseline and in conditioning pairings. While the activity of individual BLA boutons in FrA was relatively low at rest, it increased significantly upon successive pairings (*Figure 4D*). This occurred independently of the conditioning protocol and only in mice that learned (*learning+*, all CS+: 1.73 ± 0.3, all CS-: 2.07 ± 0.45, n = 5 mice; *learning-*, all CS+: 0.62 ± 0.08, all CS-: 0.77 ± 0.13, n = 5 mice; p=0.015) (*Figure 4D,E*; *Figure 4—figure supplement 1*). Interestingly, it also never occurred before the end of the first US presentation. Indeed, the number of transients observed during the first CS+ (i.e. before the delivery of the first US) was significantly lower than the other CS that were paired with the US (*learning+*, CS/No US: 1.03 ± 0.15, CS/US: 1.97 ± 0.4, n = 5 mice, p=0.011; *learning-*: CS/No US: 0.74 ± 0.05, CS/US: 0.68 ± 0.08, n = 5 mice, p=0.863) (*Figure 4E*). As a consequence, the activity of boutons measured during the first CS- was always higher than during the first CS+ (*learning+*, CS+1: 1.03 ± 0.15, CS-1: 2.03 ± 0.4, n = 5 mice, p=0.003; *learning-*, CS+1: 0.74 ± 0.05, CS-1: 0.88 ± 0.09, n = 5 mice, p=0.425) (arrow in *Figure 4E*). These data suggest that neither the tone alone nor the foot shock influence the activity of BLA-to-FrA axons. In agreement, calcium transients between conditioning trials were not modulated by the presence of CS- (*Figure 4—figure supplement 1*). In addition, BLA-to-FrA axons were never activated by auditory stimulations in naive mice (i.e. before fear conditioning) (*Figure 4—figure supplement 1*). Together, our data support the idea that BLA axons projecting to the FrA convey information about learning, that is the CS+/US association itself rather than about the nature of the auditory tones.

Then, we summed the amplitude of all calcium transients detected during each CS presentation (cumulative $\Delta F/F_0$ averaged across all CS+ or CS-). The averaged cumulative $\Delta F/F_0$ measured during CS- was always higher than during CS+ (CS-: 0.27 ± 0.07 $\Delta F/F_0$, CS+: 0.21 ± 0.06 $\Delta F/F_0$, n = 10 mice; p=0.012), revealing that the overall activity of BLA-to-FrA axons was stronger between conditioning trials, notably at the time when CS- occurred (*Figure 4F*). Importantly, the difference between CS+ and CS- related axonal activity ($\Delta$cumulative: $\Delta F/F_{0\ CS-}$ - $\Delta F/F_{0\ cs+}$) was significantly higher in mice that learned the association than in non-learners (*learning+*: 0.1 ± 0.02 $\Delta$cumulative, n = 5 mice; *learning-*: 0.01 ± 0.008 $\Delta$cumulative, n = 5 mice; p=0.013, *t-test*) (*Figure 4F*). In fact, when plotted as a function of freezing percentage observed during recall, the $\Delta$cumulative correlated positively with learning performance ($r^2$=0.89, p<0.001) (*Figure 4G*), suggesting that the level of activity of BLA-to-FrA axons during CS- is critical for the acquisition of fear memories.

## Non-linear interaction in FrA L2/3 pyramidal neurons between segregated BLA and auditory inputs

The activation of BLA neurons instructs prefrontal circuits during learning and memory recall (*Klavir et al., 2017*; *Nakayama et al., 2015*; *Stujenske et al., 2014*). However, the optical activation of the BLA alone is not sufficient to produce learned associations (*Johansen et al., 2010*). Therefore, we hypothesized that the activation of BLA axons during CS- (*Figure 4*), along with the synaptic non-linearities evoked by *WGN* (*Figures 1* and *2*), could gate fear learning (*Figure 3*) by controlling L2/3 FrA pyramidal neurons through their projections into L1.

To test this hypothesis, we first addressed the properties of BLA-to-FrA synapses in naive mice. We expressed the recombinant light-gated ion channel *channelrhodopsin*-2-YFP (ChR2; AAV9-CamKIIa-hChR2-eYFP) in the BLA and performed intracellular recordings in L2/3 FrA neurons from naive mice (*Figure 5*). BLA neurons expressing ChR2 projected to the superficial layer 1 of the ipsilateral FrA (<150 µm from the pia) (*Figure 5A*; *Figure 5—figure supplement 1A–C*), thereby most likely contacting dendritic tufts of L2/3 pyramidal neurons. Local photostimulation of ChR2-BLA axons in acute slices produced excitatory postsynaptic current (EPSC) in FrA pyramidal neurons with short

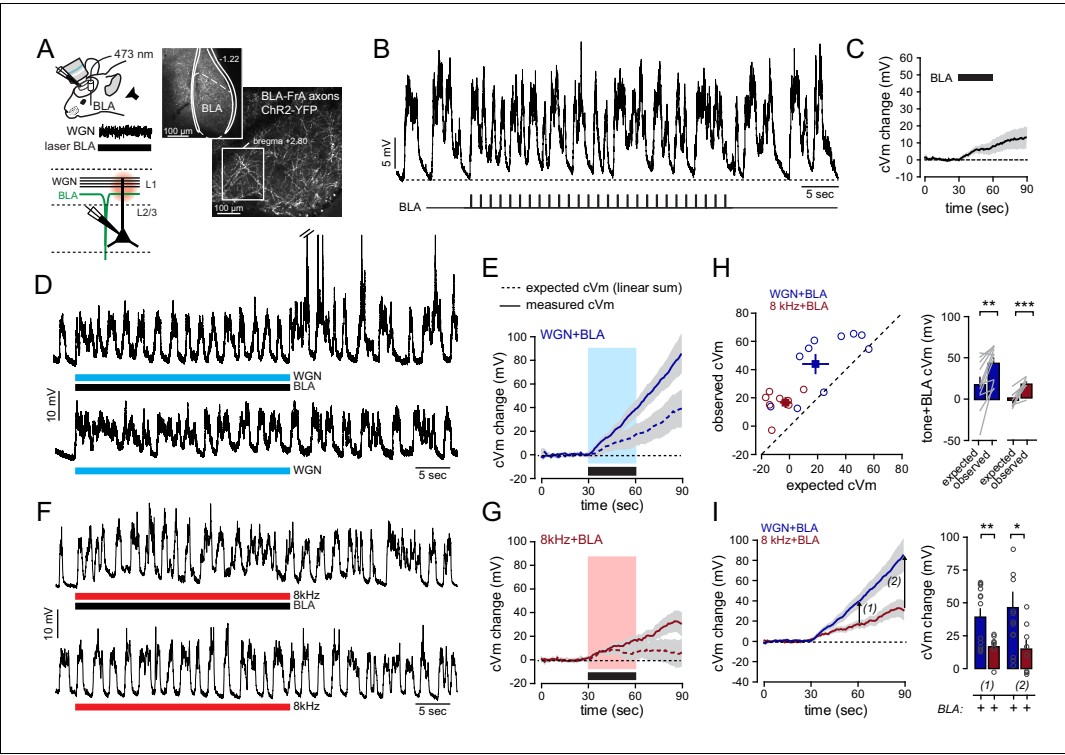

**Figure 5.** The activation of BLA-to-FrA axons supports the non-linear integration of auditory tones. (A) *Left,* co-activation protocol. ChR2-expressing BLA neurons were photo-stimulated during auditory stimulation. *Right*, representative example of the ChR2-GFP expression profile in the mouse BLA and FrA (ChR2-expressing axons were imaged in superficial layer 1 with 2P microscopy before recording). (B) Single-cell example of depolarizations in FrA by the rhythmic photostimulation of BLA neurons (27 square stimulations, 50 ms, 0.9 Hz for 30 s). (C) Averaged cVm change (±sem) evoked by the photostimulation of BLA neurons. Photo-stimulation is depicted by the black bar. (D) Example of traces of postsynaptic membrane potential recorded from individual FrA L2/3 pyramidal neurons upon *WGN* paired (*top*) or not (*bottom*) with the photo-stimulation of ChR2-expressing BLA neurons. Black and blue bars below the traces indicate the duration of the stimulation. (E) Averaged cVm change (±sem) observed upon paired stimulation (solid line) or expected from the arithmetic sum of individual depolarizations evoked by the stimulation of BLA or auditory tones alone (dotted line). Blue and black bars, auditory and BLA stimulations, respectively. (F, G) Same representation as in (D, E) but for *8 kHz.* (H) *Left*, relation between observed and expected cVm. Circles, individual cells, squares, mean ±sem. *Right*, Averaged cVm (±sem). **, p<0.01; ****, p<0.001. Gray lines indicate paired experiments. (I) *Left,* averaged observed cVm change (±sem). Arrows, analysis time points 1 (end of stimulation) and 2 (30 s later). *Right*, effect of photostimulation (+) on *WGN* (dark blue) and *8 kHz* (black red)-evoked cVm at time points 1 and 2. **, p=0.003; *, p=0.035. Circles, individual cells.

The online version of this article includes the following source data and figure supplement(s) for figure 5:

**Source data 1.** Sound and BLA-evoked depolarization.

**Figure supplement 1.** BLA-mediated synaptic inputs onto FrA L2/3 pyramidal neurons.

**Figure supplement 2.** The activation of BLA-to-FrA axons supports the non-linear integration of auditory tones.

**Figure supplement 2—source data 1.** Sound and BLA-evoked depolarization.

latencies (3.5 ± 0.36 ms, n = 9 cells) and low jitter (0.289 ± 0.04 ms, n = 9 cells), suggesting that a fraction of BLA neurons are connected monosynaptically to L2/3 FrA pyramidal neurons (*Figure 5—figure supplement 1D,E*; *Klavir et al., 2017*). In vivo, the photostimulation of BLA neurons with an implanted optical fiber produced plateau-like depolarizations in all FrA neurons (averaged peak amplitude: 6.2 ± 1.2 mV*sec, full-width at half-max (fwhm): 551 ± 80 ms; n = 13 cells). However, BLA-to-FrA inputs were mostly undetectable and became visible only when the stimulation was delivered during down states (*Figure 5—figure supplement 1F,G*), suggesting that the responses of FrA neurons to BLA stimulation in vivo are likely to be altered by on-going spontaneous activity (*Ferezou and Deneux, 2017*). When detected, the distribution of amplitudes across cells suggested

that, on average, they were highly variable (range of amplitude: 0.4 mV-14.5 mV) (*Figure 5—figure supplement 1G*). The rhythmic stimulation of ChR2-expressing BLA neurons at 0.9 Hz for 30 s (27 square pulses, 50 ms) (*Figure 5B,C*), a protocol that mimicked the pattern of auditory stimuli, generated modest cumulative depolarization (8 ± 4 mV; n = 21 cells). Taken together, our data indicate that BLA-to-FrA synapses are likely to be weak and unreliable.

We next investigated the effect of BLA activation during auditory stimulation on L2/3 FrA pyramidal neurons (*Figure 5D–I*). We first verified that *WGN* alone was able to activate FrA pyramidal neurons in mice chronically implanted with optical fibers. Similar to the effect of auditory stimulation in non-implanted mice (*Figure 5—figure supplement 2A*), *WGN* but not *8 kHz* evoked a long-lasting subthreshold depolarization (*WGN*: 18.9 ± 7 mV, n = 11 cells; *8 kHz*: −9.7 ± 3 mV, n = 8 cells; p=0.005; *t-test*). The coincident photo-activation of BLA neurons during the presentation of *WGN* did not generate somatic action potential. However, it resulted in somatic responses (observed cVm) that were significantly higher than the arithmetic sum (expected cVm) of individual depolarizations evoked by the stimulation of BLA or auditory tones alone (observed cVm: 44 ± 6.4 mV, expected cVm: 18 ± 8.8 mV, n = 11 cells; p=0.002, *paired t-test*) (*Figure 5D,E,H*; *Figure 5—figure supplement 2B,C*). We then plotted the observed vs. expected cVm change and found that the observed cVm exceeded the expected cVm, indicating that the interaction in FrA neurons between BLA and *WGN*-related inputs was clearly supralinear (*Spruston and Kath, 2004*; *Tran-Van-Minh et al., 2015*; *Figure 5H*). Supra-linear operations have been shown to depend on active dendritic conductances (*Spruston and Kath, 2004*; *Tran-Van-Minh et al., 2015*). Accordingly, we found that the application of dAP5 (1 mM) to the cortical surface blocked the effect of BLA activation during *WGN* (BLA +*WGN*: 39 ± 6.2 mV, n = 13 cells; BLA+*WGN*/+dAP5: −26.6 ± 6.7 mV, n = 7 cells; p<0.001; *t-test*), indicating that NMDARs are involved in the supra-linear integration in FrA neurons (*Figure 5—figure supplement 2F*). The photo-activation of BLA also significantly affected the cumulative potential evoked by *8 kHz* (*8 kHz*: 9.7 ± 3.3 mV vs. *8 kHz*+BLA: 16.6 ± 3.1 mV; n = 8 cells; p<0.001; *paired t-test*) (*Figure 5F,G*; *Figure 5—figure supplement 2D,E*). Although supra-linear (observed cVm: 16.6 ± 3.1 mV, expected cVm: −2.4 ± 3.3 mV, n = 8 cells; p<0.001, *paired t-test*), the BLA+*8 kHz* integration remained significantly lower than the supra-linearity generated by BLA+*WGN* (observed cVm; *WGN*: 44 ± 6.4 mV, n = 11 cells; *8 kHz:* 16.6 ± 3.1 mV, n = 8 cells; p=0.003, *t-test*). This indicates that BLA axons are necessary to produce non-linear integration of auditory inputs in FrA neurons, which is stronger for Gaussian noise than for pure tones.

## BLA-to-FrA non-linear integration of Gaussian noise gates fear learning

We next examined whether this BLA-mediated, non-linear integration of auditory inputs in FrA pyramidal neurons could play a role in the acquisition of fear memories. This question was addressed by silencing specifically the BLA neurons that project to the FrA axons during conditioning with optogenetics (*Figure 6*; *Figure 6—figure supplement 1*). Mice were injected bilaterally with a retrograde Cav-2-CMV-Cre (*Hnasko et al., 2006*) into the FrA together with either AAV9. CBA.Flex.ArchT.GFP (ArchT-expressing mice, n = 24) or AAV9.CAG.Flex.eGFP (control GFP-expressing mice, n = 25) into BLA bilaterally (*Figure 6A*). This resulted in the restricted expression of the light-driven inhibitory proton pump ArchT (or GFP for controls) in a target-specific fraction of BLA neurons that project to the FrA (*Figure 6B–D*). In contrast to BLA neurons that project directly to the mPFC (*Hoover and Vertes, 2007*; *Likhtik and Paz, 2015*; *Senn et al., 2014*), BLA-to-FrA axons do no send collaterals to the prelimbic and infralimbic subdivisions of the mPFC (*Figure 6C,D*).

Mice were then submitted to auditory fear conditioning, and we analyzed the impact of opto-stimulation on freezing behaviors for each counter-balanced protocol (*Figure 6—figure supplement 1*). We found that the time-locked suppression of BLA-to-FrA communication during negatively paired *WGN* (CS-, protocol 1b) significantly decreased the fraction of ArchT-mice that learned the association (GFP: 87.5%; ArchT: 22%, p=0.005, *Pearson's $\chi^2$ test*) and freezing behaviors upon subsequent CS+ (*8 kHz*) presentation (GFP: 55.9 ± 8%, n = 8 mice; ArchT: 34.9 ± 3%, n = 7 mice; p=0.037, *t-test*) (*Figure 6E,F*; *Figure 6—figure supplement 1*). These results were similar to those obtained when FrA neurons were inhibited during conditioning (*Figure 3*; *Figure 6—figure supplement 1*), thus confirming that negatively paired *WGN* participates in the formation of fear traces. In contrast, neither the fraction of mice that learned (GFP: 78%; ArchT: 44%, p=0.145, *Pearson's $\chi^2$ test*) nor freezing responses during recall (CS+; GFP: 34.8 ± 3.9, n = 9 mice; ArchT: 29.3 ± 8, n = 9 mice; p=0.530, *t-test*) were affected by blocking the activity of BLA-to-FrA axons

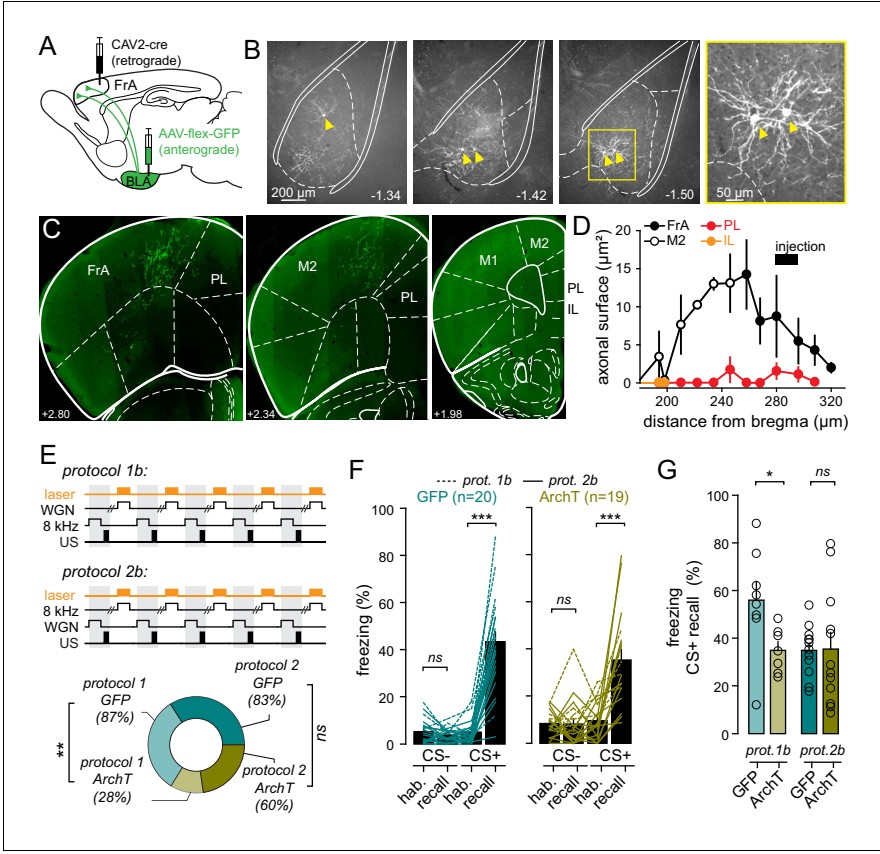

**Figure 6.** BLA-to-FrA projecting neurons are required for auditory fear learning when WGN is used as a CS-. (**A**) Experimental strategy. Mice were infected with AAV9.CBA.Flex.ArchT-GFP (or AAV9.CAG.Flex.eGFP for controls) in BLA and CAV2.CMV.CRE in FrA, and chronically implanted with optical fibers in both BLA. (**B**) Neurons expressing GFP were found throughout the entire BLA, but not the LA. (**C**) Example of expression profile of GFP in the FrA and the M2. Note the absence of GFP-expressing axons in the prelimbic (PL) and infralimbic (IL) subdivisions of the mPFC. (**D**) Distribution of GFP-expressing axons; in different frontal structure; as a function of the distance from the bregma. (**E**) *Top*, experimental fear-conditioning protocols. FrA neurons expressing ArchT (or GFP) were photo-stimulated during the presentation of CS- (*WGN* for protocol 1b or *8 kHz* for protocol 2b). US, unconditional stimulus (footshock). *Bottom*, fraction of mice that learned the cue-shocked association for each protocol. (**F**) Effect of light during learning on freezing behaviors during recall as compared to habituation (hab.) in GFP-expressing mice (left; n = 20) and ArchT-expressing mice (right, n = 19). Turquoise and khaki lines represent individual mice. (**G**) Freezing behaviors quantified during CS+ in GFP- and ArchT-expressing mice (p=0.235). Circles, individual mice. ***, p<0.001; **, p<0.01; ns, non-significant.

The online version of this article includes the following source data and figure supplement(s) for figure 6:

**Source data 1.** Freezing responses.

**Figure supplement 1.** Fear-conditioning protocol 1: comparisons between behavioral conditions.

during negatively-paired *8 kHz* (CS-, protocol 2b) (*Figure 6E,G*). Similar results were obtained when photo-stimulation was administered during CS+ (*WGN*, protocol 2a).

Altogether, our data support the idea that if Gaussian noise is presented when the activity of BLA neurons is the strongest (i.e. between conditioning trials, *Figure 4*) it facilitates discriminative learning. In support of this hypothesis, fear conditioning with protocol one produced more freezing responses in control mice than with protocol 2 (*Figure 6—figure supplement 1*). In addition, CS+ evoked freezing responses were significantly lower during recall in mice that were conditioned in the absence of CS- (*Figure 6—figure supplement 1*).

## Discussion

The present study investigates the role of the BLA-to-FrA circuit in the integration of auditory cues, and how this process participates in the acquisition of fear traces during conditioning. Taken together, our data demonstrate that rather than being an obstacle, Gaussian noise facilitates auditory fear learning when not paired to the foot shock (i.e. delivered between conditioning trials), thereby confirming the sophisticated nature of differential conditioning protocols (*Grosso et al., 2015*; *Hall, 2002*; *McDonnell and Abbott, 2009*). This is possibly due to: (1) the specific dendritic and somatic responses of FrA pyramidal neurons to *WGN*;(2) the activation of BLA-to-FrA axons between conditioning pairings, which might support (3) the non-linear integration of Gaussian noise in FrA neurons. Unlike *WGN*, none of the optogenetic manipulations aiming at altering pure tone processing in FrA and BLA during conditioning had an effect on fear learning. Thus, our data bring additional perspectives to central questions regarding how frontal circuits contribute to learning, thus going beyond the BLA-mPFC interactions classically described in fear learning studies (*Likhtik and Paz, 2015*). Lesion studies revealed the decisive functions of cortical areas when animals learn to predict threat specifically from complex tones (*Grosso et al., 2015*; *LeDoux, 2000*; *Letzkus et al., 2011*; *Ohl et al., 1999*). Thus, despite the dendritic and somatic properties of WGN that we observed in the FrA, it remains likely that other complex tones, including natural tones or more complex frequency-modulated tones covering a wider spectrum of pure frequencies, could also facilitate learning in a similar way than WGN.

Accumulating evidence from anatomical and functional studies has demontrated that despite its pivotal role in the acquisition and expression of associations between sensory stimuli and the emotional valence of these stimuli (*Laviolette et al., 2005*; *Roesch et al., 2010*), the mPFC is not directly involved in sensory processing (*Hoover and Vertes, 2007*; *Martin-Cortecero and Nuñez, 2016*; *Zhang et al., 2016*). In contrast, due to its anatomical connections with distributed cortical and subcortical regions (*Hoover and Vertes, 2007*; *Zhang et al., 2016*), the FrA might serve as a hub that coordinates incoming sensory information before reaching the mPFC. Here, we show that it is indeed required for fear learning in a rather unconventional way, and further clarify the underlying synaptic mechanisms. To our knowledge, this is the first demonstration that auditory sensory stimulation produces NMDAR-dependent depolarization in FrA L2/3 pyramidal neurons and activates their dendrites during both anesthesia and wakefulness. These depolarizations persisted beyond the end of the stimulation, which possibly reflects changes in the dynamics of network activity through local recurrent connections (*Ferezou and Deneux, 2017*; *Kasanetz et al., 2002*; *Shu et al., 2003*). The somatic and dendritic events were more pronounced with Gaussian noise than with pure frequency auditory stimulation. Although we cannot rule out that *WGN* tones are structured and further abstracted throughout the entire auditory system (*Deneux et al., 2016*), the simplest explanation is that frequency-tuned spines are distributed widely and heterogeneously throughout the same FrA dendrite, as already reported in the auditory cortex of anesthetized mice (*Chen et al., 2011*). As a consequence, the multiple frequencies composing *WGN* might promote the activation of a dense pattern of neighboring spines that in turn facilitate the generation and propagation of local non-linear events toward the soma (*Antic et al., 2010*). Multiple calcium transients occurring simultaneously in multiple dendritic branches are necessary to affect somatic voltage (*Palmer et al., 2014*). Both those findings and ours suggest that *WGN*-induced depolarization is the consequence of multiple calcium events that occur in different dendritic branches of the same neuron. In contrast, pure-frequency tone appeared unable to activate enough branches simultaneously, thereby making the alteration of somatic voltage less probable. In support of this hypothesis, we found that simple linear chirps produced more somatic depolarization than pure tones. This further suggests that more complex frequency-modulated tones could theoretically also activate FrA pyramidal neurons and affect fear learning like WGN did in our experimental conditions.

The FrA and the BLA are strongly interconnected (*Lai et al., 2012*; *Mátyás et al., 2014*; *Nakayama et al., 2015*). However, the functional properties of these connections remain unknown, notably during learning. Our results confirm that BLA neurons project to the superficial layer of the FrA, thereby most likely contacting dendrites of L2/3 pyramidal neurons. However, and in contrast to what we observed in vitro, the photo-stimulation of ChR2-expressing BLA neurons in vivo at the same intensity than for behavioral experiments produced depolarizations in FrA pyramidal neurons that were rather weak and unreliable. The difference between in vitro and in vivo conditions may

reflect the interaction between evoked responses and on-going spontaneous activity that occurs in vivo (*Ferezou and Deneux, 2017*). It thus seems unlikely that they can summate during rhythmic activation to create favorable conditions for the integration of coincident sensory-driven inputs. Alternatively, it is possible that the modest activation of BLA synapses in FrA apical dendrites gates the propagation towards the soma of tone-evoked dendritic events which could enhance the generation of somatic action potential (*Jarsky et al., 2005*; *Palmer et al., 2014*). Here, we observed that the coincident activation of BLA-to-FrA inputs increased both *WGN* and *8 kHz*-evoked depolarization non-linearly. Although the BLA+*WGN* nonlinearities were much stronger than those generated by BLA+*8 kHz*, it did not generate somatic action potential in FrA pyramidal neurons. While the absence of spiking could have been caused by the anesthesia, it is known that the non-linear interaction between compartmentalized streams of neural activity also induces long-lasting changes in synaptic strength and intrinsic excitability (*Dudman et al., 2007*; *Gambino et al., 2014*; *Jarsky et al., 2005*; *Larkum, 2013*; *McGaugh, 2013*; *Xu et al., 2012*). In agreement, we found that the activation of BLA neurons during *WGN* affected subsequent spontaneous activity more robustly and for a longer period of time than during *8 kHz*. Although this effect was analyzed no longer than 30 s after the end of the stimulation, it also prompts the speculation that the alteration of FrA membrane potential evoked by the BLA+*WGN* nonlinearities might reflect a global brain state change which would modify the integration of future incoming sensory inputs, such as it occurs during consecutive pairing upon conditioning.

The mechanism by which these BLA-mediated nonlinearities affect learning remains unclear. The BLA presumably transfers information to the FrA that is relevant for fear learning (*Lai et al., 2012*; *Nakayama et al., 2015*). Surprisingly, inhibiting the activity of FrA neurons during CS-, or BLA neurons projecting to the FrA, attenuated freezing responses in response to CS+ during recall. However, this counterintuitive effect occurred only when *WGN*, but not *8 kHz* tone, was used as CS- during conditioning (protocol 1). This alteration of learning is unlikely to be the consequence of insufficient activation of the BLA. Instead, given the low number of BLA neurons expressing ArchT and their precise inhibition during CS-, an alternative explanation is that when *WGN* is combined with the activation of BLA projecting axons, which is maximal between conditioning trials, it might promote the representation of sensory cues predicting threat (i.e. CS+, *8 kHz*) within the FrA (*Hall, 2002*). This hypothesis is supported by the late modification of membrane potential fluctuations observed after the activation of BLA with *WGN* presentation, which occurred when an *8 kHz* tone was presented during conditioning. Interestingly, omitting CS-/WGN during fear conditioning does not prevent learning. However, freezing responses during recall were significantly lower than when CS- were presented between conditioning trials, but similar to those obtained when FrA or BLA-to-FrA neurons were inhibited during CS. Taken together, it indicates that the BLA-mediated, non-linear integration of *WGN* in FrA neurons between conditioning trials facilitates the acquisition of fear memory traces.

BLA neurons send multiple projections to cortical and subcortical areas that have been shown to project also to the FrA. It is thus possible that the information is transmitted from the BLA to the FrA through an indirect pathway (*Nakayama et al., 2015*; *Price, 2003*). Here, we found that BLA-to-FrA neurons do not send axonal collaterals to the mPFC. In addition, we demonstrate that the expression of the genetically encoded calcium indicator GCaMP6 in BLA neurons makes it possible to monitor optically the activity of target-specific BLA axons during learning. Using this strategy in awake mice, we demonstrate for the first time that BLA-to-FrA axons are progressively recruited upon successive conditioning trials, thereby ruling out the indirect activation of the FrA circuit. First, we show that BLA-to-FrA axonal activity is never affected by the presentation of auditory cues alone. Our data contrast with previous work showing an increase in local field potential and unit activity in the BLA upon auditory stimulation (*Collins and Paré, 2000*), and suggest instead the existence of a subpopulation of BLA neurons projecting specifically to the FrA that might play a specific role during the learning of emotion. Although it remains unknown whether BLA axons might convey contextual information to the FrA, they seem to transmit integrated information about the association itself. First, these axons are activated only after the first CS+/US pairing. Then, the level of activity of BLA-to-FrA axons was stronger between conditioning trials (i.e. during CS- presentation) and correlated with learning performance. This suggests a putative *Hebbian*-like frontal mechanism that could integrate any Gaussian noise that is contiguous to the maximal activation of BLA-to-FrA axons (*Johansen et al., 2010*; *Johansen et al., 2014*; *Larkum, 2013*; *Nakayama et al., 2015*). Supporting

this idea, we found that *WGN* but not *8 kHz*-dendritic and somatic signaling were reduced in conditioned mice in a learning-dependent manner, suggesting that NMDAR-dependent plasticity mechanisms in FrA neurons were engaged during fear learning. While this mechanism would occur only if *WGN* is presented between conditioning trials (and thus used as CS-), it might eventually facilitate the recruitment of neurons into specific cue memory traces. Dendritic plateau potentials have been shown to regulate synaptic strength and synaptic plasticity (*Cichon and Gan, 2015*; *Du et al., 2017*; *Gambino et al., 2014*; *Humeau and Lüthi, 2007*; *Palmer et al., 2014*), which might subsequently facilitate the stabilization or pruning of synaptic inputs during learning (*Holtmaat and Caroni, 2016*; *Li et al., 2017*). In support of the latter, the level of fear learning has been shown to correlate with the percentage of spine elimination in FrA (*Lai et al., 2012*) which possibly explains the negative relation we observed during anesthesia between *WGN*/(CS-)-evoked subthreshold depolarizations and the strength of learning.

Collectively, our data reveal the specific properties of Gaussian noise in FrA during fear conditioning. The question arises as to the function or benefit of Gaussian noise in the BLA-to-FrA circuit during learning. Previous studies highlighted the critical function of the BLA in attention for learning (*Laviolette et al., 2005*; *Roesch et al., 2010*). Here, we show that the activation of BLA-to-FrA axons is independent of the nature of the CS presented. It thus seems unlikely that BLA-to-FrA axons convey the emotional valence of this association. In addition, the activation of BLA alone, while necessary, is not sufficient to trigger learning (*Johansen et al., 2010*; *Johansen et al., 2014*). BLA neurons might signal to frontal circuits any new association independently of its valence, which might subsequently be assigned by the mPFC (*Klavir et al., 2013*; *Likhtik and Paz, 2015*). While acoustic noise is often viewed as a disturbing variable, it can enhance signal processing, facilitate sensory signaling and improve cognitive performance, notably in individuals with poor attention (*McDonnell and Abbott, 2009*; *McDonnell and Ward, 2011*; *Stein et al., 2005*). Therefore, given that noise is abundant in the environment and communication of most mammals, it might facilitate learning in coordination with BLA-to-FrA inputs.

## Materials and methods

All experiments were performed in accordance with the Guide for the Care and Use of Laboratory Animals (National Research Council Committee (2011): Guide for the Care and Use of Laboratory Animals, 8th ed. Washington, DC: The National Academic Press.) and the European Communities Council Directive of September 22th 2010 (2010/63/EU, 74). Experimental protocols were approved by the institutional ethical committee guidelines for animal research (N°50DIR_15-A) and by the French Ministry of Research (N°02169.01). We used male C57Bl6/J 6 weeks old mice from Charles River that were housed with littermates (3–4 mice per cage) in a 12 hr light-dark cycle. Cages were enriched and food and water were provided ad libitum.

### Auditory tones

The different sounds used in this study are all composed of 27 pips of 50 ms presented at 0.9 Hz, regardless of their spectral properties. Each 50 ms-long pips (sampling frequency: 44100 Hz) were generated by producing 2205 samples (i) following a Gaussian distribution for WGN, (ii) of a sinusoidal signal for pure tones, and (iii) of a swept-frequency cosine signal for linear chirp tones. Pure and Chirp pips were created by using the 'sin' and 'chirp' functions in Matlab, respectively. WGN was created by using the 'randn' function in Matlab, which generates random numbers that follow a Gaussian distribution. WGN is thus a true random signal with equal intensity at all frequencies. Auditory tones were further low-pass filtered at 20 kHz to avoid aliasing, but also to limit ultrasonic frequencies, which could trigger innate fear-induced defensive responses in rodents, such as escape of freezing, without associative learning (*Brudzynski and Chiu, 1995*).

### Surgery and virus injection

Mice were anesthetized with an intraperitoneal (i.p.) injection of a mix containing medetomidine (sededorm, 0.27 mg kg$^{-1}$), midazolam (5 mg kg$^{-1}$), and fentanyl (0.05 mg kg$^{-1}$) in sterile NaCl 0.9% (MMF-mix). Analgesia was achieved by local application of 100 µl of lidocaine (lurocaine, 1%) and subcutaneous (s.c.) injection of buprenorphine (buprécare, 0.05 mg kg$^{-1}$). 40 µl of dexamethasone (dexadreson, 0.1 mg ml$^{-1}$) was administrated intramuscularly (i.m.) in the quadriceps to prevent

inflammation potentially caused by the friction of the drilling. A heating-pad was positioned underneath the animal to keep the body temperature at 37°C. Eye dehydration was prevented by topical application of ophthalmic gel. The skin above the skull was disinfected with modified ethanol 70% and betadine before an incision was made. Stereotaxic injections were done as previously described (*Gambino et al., 2014*). Briefly, the bregma and lambda were aligned (x and z) and a hole for injection was made using a pneumatic dental drill (BienAir Medical Technologies, AP-S001). The injections were targeted either to the layer 2/3 of the FrA (from bregma: AP, +2.8 mm; DV, −0.2–0.3 mm; ML ± 1.0 mm) or to the BLA (from bregma: AP, −1.3 mm; DV, −4.5 to 4.8 mm; ML,±2.9 mm), or to both at the same time. A total of 200 nl of virus were injected at a maximum rate of 60 nl/min, using a glass pipette (Wiretrol, Drummond) attached to an oil hydraulic manipulator (MO-10, Narishige).

The following viruses were used depending on the experiments. AAV-ChR2 (AAV9.CamKIIa. hChR2(H134R).eYFP.WPRW.SV40, Penn Vector Core) was unilaterally injected in the right BLA, whereas AAV-ArchT-Flex (AAV9.CBA.flex.Arch-GFP.WPRE.SV40, Penn Vector Core) and CAV2-Cre (Cav2.CMV.Cre, IGMM BioCampus Montpellier) were bilaterally injected into the BLA and FrA, respectively. Control experiments were performed using an AAV containing the DNA construct for GFP (AAV9.CAG.flex.eGFP.WPRE.bGH). For axonal calcium imaging, AAV-GCaMP6f (AAV1.Syn. GCaMP6f.WPRE.SV40, Penn Vector Core) was injected to the right BLA. For dendritic calcium imaging, AAV-GCaMP6s (AAV9.Syn.Flex.GCaMP6s.WPRE.SV40, Penn Vector Core) and a 1:10,000 dilution of AAV-Cre (AAV1.hSyn.Cre.WPRE.hGH, Penn Vector Core) were injected together into the right FrA. After injections, the viruses were allowed to diffuse for at least 10 min before the pipette was withdrawn. Mice were then either prepared for cranial window implantation or waked-up by a sub-cutaneous injection of a mixture containing atipamezole (revertor, 2.5 mg kg$^{-1}$), flumazenil (0.5 mg kg$^{-1}$), and buprenorphine (buprécare, 0.1 mg kg-1) in sterile NaCl 0.9% (AFB-mix).

The cranial windows were made as previously described (*Gambino et al., 2014*). Briefly, after skull's exposure a ⌣ 5 mm plastic chamber was attached on the area of interest and a 3 mm craniotomy was made on the right hemisphere above FrA and M2, with a pneumatic dental drill, leaving the dura intact. The craniotomy was covered with sterile saline (0.9% NaCl) and sealed with a 3-mm glass cover slip after viral injection (for imaging experiments). The chamber, the cover slip and a custom-made stainless steel head stage were well attached to the skull using dental acrylic and dental cement (Jet Repair Acrylic, Lang Dental Manufacturing).

To evaluate the viral expression profiles in BLA and FrA, fixed brain slices were imaged post-hoc using a wide-field epifluorescence microscope (Nikon, Eclipse N-iU). Illumination was set such that the full dynamic range of the 16-bit images was utilized. A two-dimensional graph of the intensities of pixel was plot using Fiji Software. 16-bit images' brightness was processed and masks were registered to the corresponding coronal plates (ranging from −1.94 to −2.70 mm) of the mouse brain atlas using Illustrator (Adobe), at various distances anterior (FrA) or posterior (BLA) to the bregma.

## Fear conditioning and quantification of learning

At least 5 days before starting behavioral experiments, mice went through handling with the same experimenter that performed the experiments in order to decrease stress. For consistency across experiments, mice were then habituated to auditory tones during three successive days. During habituation, mice were placed on the conditioning compartment (context A, consisting of a squared box with a grid floor that allows the delivery of a foot shock and with home cage litter under; cleaned between individuals with 70% ethanol). Two conditional auditory stimuli (CS) (8 kHz pure tone; and white Gaussian noise (WGN); each composed of 27 pips, 50 ms in duration, 0.9 Hz for 30 s) were presented four times with a 80 dB sound pressure level and variable inter stimulus interval (ISI). The freezing time during each CS presentation was measured and the mice returned to their home cage. Mice were fear conditioned 24 hr after the last habituation phase by using a classical differential protocol. Briefly, mice were exposed to context A and five auditory tones (CS+) were paired with the unconditional stimulus (US, 1 s foot-shock, 0.6 mA). The onset of US coincided with the CS+ offset. 5 CS- presentations were intermingled with CS+ presentations with a variable (10–60 s) ISI. CS were counterbalanced with WGN and 8 kHz pure tones being used as CS+ and CS-, respectively. Both CS+ and CS- have always the same duration (30 s each) regardless of their spectral properties, with the same number of pips (27 pips at 0.9 Hz). Only the delays between the end of a pairing (CS +/footshock) and the beginning of the next CS-, as well as the delays between the end of a CS- and

the beginning of the next CS+ have been chosen randomly during conditioning. Recall tests were carried out 24, 48, and 72 hr after the conditioning phase by measuring the freezing time during the presentation of 2 CS+ and 2 CS- in a new context (context B, consisting of a cylindrical white compartment with home cage litter on the floor; cleaned between individuals with septanios MD 2%).

For optogenetic experiments using *archeorhodopsin* (ArchT) or GFP controls, mice were subjected to the same behavioral protocol described above. Optogenetic inhibition of FrA neurons or BLA-to-FrA projections upon CS presentation was achieved during the conditioning phase by synchronizing each pip (50 ms) composing the CS+ or the CS- with a 50-ms-laser pulse. For the experiments in which the conditioning phase was taken place under the two-photon microscope, the context consisted of the microscope shading box in which the mice were head-restrained in a custom tube containing a shocking grid at the bottom. CS and US presentations were triggered by a MATLAB routine, associated to a pulse-stimulator (Master-8, A.M.P.I) capable of triggering the foot shock. For somatic and dendritic calcium imaging experiments, behavior was assessed at least 6 hr after imaging sessions. For whole-cell recordings experiments, mice were anesthetized and prepare for patch recordings immediately after behavior.

For each behavioral session, the total time duration (s) of freezing episodes upon CS+ and CS- presentation was quantified automatically using a fire-wire CCD-camera connected to an automated freezing detection software (AnyMaze, Ugo Basile, Italy), and expressed as % of freezing. Learning index was further quantified for each CS by multiplying the % of freezing in each condition by the corresponding index of discrimination by using the following equation:

$$learning\ index\ (\%) = freezing\ (\%) \times \frac{freezing\ CS^+(\%) - freezing\ CS^-(\%)}{freezing\ CS^+(\%) + freezing\ CS^-(\%)}$$

Learning index <20% during recall was considered as a failure of conditioning.

## In vivo whole cell recordings

Isoflurane (4% with ~0.5 l min$^{-1}$ O$_2$) combined with an i.p. injection of urethane (1.5 g kg$^{-1}$, in lactated ringer solution containing in [mM] 102 NaCl, 28 Na L Lactate, 4 KCl, 1.5 CaCl$_2$) was used to induce anesthesia and prolonged by supplementary urethane (0.15 g kg$^{-1}$) if necessary. To prevent risks of inflammation, brain swelling and salivary excretions, 40 µl of dexamethasone (dexadreson, 0.1 mg ml$^{-1}$, i.m.) and glycopyrrolate (Robinul-V, 0.01 mg kg$^{-1}$, s.c.) were injected before the surgery. Adequate anesthesia (absence of toe pinch and corneal reflexes, and vibrissae movements) was constantly checked and body temperature was maintained at 37°C using a heating-pad positioned underneath the animal. Ophthalmic gel was applied to prevent eye dehydration. Analgesia was provided as described for viral injection (with lidocaine and buprenorphine). After disinfection of the skin (with modified ethanol 70% and betadine), the skull was exposed and a ~3 mm plastic chamber was attached to it above the prefrontal cortex using a combination of super glue (Loctite) and dental acrylic and dental cement (Jet Repair Acrylic, Lang Dental Manufacturing). A small ~1×1 mm craniotomy centered above the FrA (+2.8 mm from bregma, ±1.0 mm midline) was made using a pneumatic dental drill, leaving the dura intact.

Whole-cell patch-clamp recordings of L2/3 pyramidal neurons were obtained as previously described (*Gambino et al., 2014*) Briefly, high-positive pressure (200–300 mbar) was applied to the pipette (5–8 MΩ) to prevent tip occlusion, when passing the pia. Immediately after, the positive pressure was reduced to prevent cortical damage. The pipette resistance was monitored in the conventional voltage clamp configuration during the descendent pathway through the cortex (until −200 µm from the surface) of 1 µm steps. When the pipette resistance abruptly increased, the 3–5 GΩ seal was obtained by decreasing the positive pressure. After break-in, Vm was measured, and dialysis was allowed to occur for at least 5 min before launching the recording protocols. Current-clamp recordings were made using a potassium-based internal solution (in mM: 135 potassium gluconate, 4 KCl, 10 HEPES, 10 Na2-phosphocreatine, 4 Mg-ATP, 0.3 Na-GTP, and 25 µM, pH adjusted to 7.25 with KOH, 285 mOsM), and acquired using a Multiclamp 700B Amplifier (Molecular Devices). Spontaneous activity was recorded prior, during, and after the presentation of auditory stimulation. Spiking pattern of patched cells was analyzed to identify pyramidal neurons. dAP5 (1 mM, Tocris) was topically applied to the dura mater, before whole cell recordings. Offline analysis was performed

using custom routines written in Sigmaplot (Systat), IGOR Pro (WaveMetrics) and Matlab (Mathworks).

## In vivo optogenetics

After virus injection for ChR2 or ArchT expression, mice were subsequently implanted with fiber optic cannula for optogenetics (CFML22U, Thorlabs) in the BLA. The optic fibers were previously cleaved with a fiber optic scribe (S90R, Thorlabs) at 4.5 mm for BLA. The cannula were guided and stereotaxically inserted inside the brain with the help of a cannula holder (XCL, Thorlabs) through the same burr hole used for the viral injections (BLA coordinates from bregma: AP, −1.3 mm; DV, −4.5 mm; ML,±2.9 mm) and secured in place with a mix of super glue (Loctite) and dental acrylic and dental cement (Jet Repair Acrylic, Lang Dental Manufacturing). Anesthesia was reversed using AFB-mix for mice assigned to behavioral experiments. For in vivo photostimulation of ChR2-expressing BLA neurons, the fiber optic cannula and the optogenetic patch cable (M83L01, Thorlabs) were connected through a ceramic split mating sleeve (ADAL1, Thorlabs). The patch cable was then coupled to a blue DPSS laser (SDL-473-050MFL, Shanghai Dream Lasers Technology) which was triggered by a pulse-stimulator (Master-9, A.M.P.I), able to synchronize 50 ms laser pulses with 50 ms sound pips composing the CS. For inhibition of BLA-to-FrA projections during learning, in vivo bilateral optic stimulation of ArchT-expressing neurons was achieved by coupling the optic fibers implanted in BLA to a multimode fiber optic coupler (FCMH2-FCL, Thorlabs), with a ceramic split mating sleeve, and subsequently connected to a yellow DPSS laser (SDL-LH-1500, Shanghai Dream Lasers Technology).

## In vitro whole-cell recordings

Mice were anesthetized with a mixture of ketamine/xylazine (100 mg/kg and 10 mg/kg respectively) and cardiac-perfused with ice-cold, oxygenated (95% $O_2$, 5% $CO_2$) cutting solution (NMDG) containing (in mM): 93 NMDG, 93 HCl, 2.5 KCl, 1.2 $NaH_2PO_4$, 30 $NaHCO_3$, 25 Glucose, 10 $MgSO_4$, 0.5 $CaCl_2$, 5 Sodium Ascorbate, 3 Sodium Pyruvate, 2 Thiourea, and 12 mM N-Acetyl-L-cysteine (pH 7.3–7.4, with osmolarity of 300–310 mOsm). Brains were rapidly removed and placed in ice-cold and oxygenated NMDG cutting solution (described above). Coronal slices (300 µm) were prepared using a Vibratome (VT1200S, Leica Microsystems, USA) and transferred to an incubation chamber held at 32°C and containing the same NMDG cutting solution. After this incubation (9–11 min), the slices were maintained at room temperature in oxygenated modified ACSF containing (mM): 92 NaCl, 2.5 KCl, 1.2 $NaH_2PO_4$, 30 $NaHCO_3$, 20 HEPES, 25 Glucose, 2 $MgSO_4$, 2 $CaCl_2$, 5 Sodium Ascorbate, 3 Sodium Pyruvate, 2 Thiourea, and 12 mM N-Acetyl-L-cysteine (pH 7.3–7.4, with osmolarity of 300–310 mOsm) until recording.

Whole-cell recordings of layer 2/3 FrA principal neurons were performed on coronal slices (from bregma: +2.58 mm to +3.08 mm) at 30–32°C in a superfusing chamber. Patch electrodes (3–5 MΩ) were pulled from borosilicate glass tubing and filled with a K-gluconate-based intracellular solution (in mM: 140 K-gluconate, 5 QX314-Cl, 10 HEPES, 10 phosphocreatine, 4 Mg-ATP and 0.3 Na-GTP) (pH adjusted to 7.25 with KOH, 295 mOsm). BLA-to-FrA monosynaptic EPSCs were elicited by 1–50 ms light stimulations delivered by an ultrahigh power 460 nm LED (Prizmatix Ltd, Israel). Data were recorded with a Multiclamp700B (Molecular Devices, USA), filtered at 2 kHz and digitized at 10 kHz. Data were acquired and analysed with pClamp10.2 (Molecular Devices).

## Two-photon laser-scanning microscope (2PSLM)-based calcium imaging

Head-fixed awake mice were placed and trained under the microscope every day for at least 7 days prior to the experiment, and then imaged 21 to 35 days after virus injection using an in vivo non-descanned FemtoSmart 2PLSM (Femtonics, Budapest, Hungary) equipped with a × 16 objective (0.8 NA, Nikon). The MES Software (MES v.4.6; Femtonics, Budapest, Hungary) was used to control the microscope, the acquisition parameters, and the TTL-driven synchronization between the acquisition and auditory/footshock stimuli. The GCaMPs were excited using a Ti:sapphire laser operating at λ = 910 nm (Mai Tai DeepSee, Spectra-Physics) with an average excitation power at the focal point lower than 50 mW. Time-series images were acquired within a field-of-view of 300 × 300 µm (256 lines, 1 ms/line) for axons; for dendrite: 200 × 60 µm (64 lines, 0.5 ms/line). Each imaging session consisted of 30 s of baseline recording followed by eight Gaussian and eight pure (8 kHz)-tone

auditory stimuli delivered with pseudo-random delays. We imaged on average 3500 frames (~900 s) per session, and no visible photo-bleaching was observed. Images were then analyzed as previously described (*Gambino et al., 2014*) using Fiji and Matlab (Mathworks). We registered images over time and corrected XY motion artifacts within a single imaging session by using cross-correlation based on rigid body translation (Stack aligner, Image J, NIH, USA). Motion corrections were then assessed by computing pair-wise 2D correlation coefficient (Image correlation, Image J, NIH, USA), and frames were discarded from the analysis if lower than 0.7. Regions of interest (ROIs) putative axonal boutons were selected and drawn manually. All pixels within each ROI were first averaged providing a single time-series of raw fluorescence. To limit the effect of fluorescence drift over time, the baseline fluorescence ($F_0$) was calculated as the mean of the lower 50% of previous 3 s fluorescence values. Change in fluorescence ($\Delta F_t/F_0$) was defined as $(F_t-F_0)/F_0$, were $F_t$ is the fluorescence intensity at time $t$ (time of the first pixel in each frame). Calcium events were then detected using a template-based method with a custom library of calcium transients. Templates were created by extracting and averaging segments of data that were visually identified as corresponding to a transient. Calcium transients whose peak amplitude reached a 3 X background standard deviation threshold were further considered for analysis. Each detected event was inspected visually and analysis was restricted to detected events rather than on raw fluorescence. For extracting spatial profiles of dendritic calcium events, small ROIs of 2 × 2 pixels are generated along the dendrite in Fiji. The spread of $Ca^{2+}$ events was then quantified by calculating the full-width at half-max (fwhm, expressed as % of total dendritic length) of the normalized Gaussian fit at the time when the averaged $\Delta F/F_0$ was maximal.

## Statistics

Experiments and analysis were conducted blind to the operator. Data are presented as the median ±interquartile range or mean ±sem (except where stated differently). All statistics were performed using Matlab (Mathworks) and Sigmaplot (Systat) with an α significant level set at 0.05. Normality of all value distributions and the equality of variance between different distributions were first assessed by the Shapiro-Wilk and Levene median tests, respectively. Standard parametric tests were only used when data passed the normality and equal variance tests. Non-parametric tests were used otherwise. Only two-sided tests were used. When applicable, pair-wise multiple post-hoc comparisons were done by using the Holm-Sidak method. No statistical methods were used to estimate sample size, but β-power values were calculated for parametric tests.

## Acknowledgements

We thank S Deforges, E Normand, and B Darracq (Imetronic) for their technical expertise and support, and A Holtmaat and S Valerio (AquiNeuro) for their critical reading of our manuscript, and all the members of the Gambino laboratory for technical assistance and helpful discussions. We thank K Deisseroth and Stanford University, E Boyden and MIT, EJ Kremmer and the IGMM BioCampus Montpellier, LL Looger and D Kim from the GENIE project, and K Svoboda at the Janelia Farm Research Campus (HHMI) for distributing viral vectors. This project has received funding from (to FG): the European Research Council (ERC) under the European Union's Horizon 2020 research and innovation program (NEUROGOAL, grant agreement n° 677878), the FP7 Marie-Curie Career Integration program (grant agreement n° 631044); the ANR JCJC (grant agreement n° 14-CE13-0012-01), the University of Bordeaux (Initiative of Excellence senior chair 2014), the Laboratory of Excellence (LabEx) Brain grant 2015, and from the Region Nouvelle Aquitaine. NC is supported by a Marie Skłodowska-Curie individual fellowship under the European Union's Horizon 2020 research and innovation program (AXO-MATH; grant agreement n° 798326).

## Additional information

### Funding

| Funder | Grant reference number | Author |
| --- | --- | --- |
| H2020 European Research Council | 677878 | Frederic Gambino |

| | | |
|---|---|---|
| FP7 People: Marie-Curie Actions | 631044 | Frederic Gambino |
| H2020 Marie Skłodowska-Curie Actions | 798326 | Nicolas Chenouard |
| ANR | 14-CE13-0012-01 | Frederic Gambino |
| University of Bordeaux | Initiative of Excellence senior chair 2014 | Frederic Gambino |
| Labex | Brain grant 2015 | Frederic Gambino |

The funders had no role in study design, data collection and interpretation, or the decision to submit the work for publication.

### Author contributions
Mattia Aime, Data curation, Formal analysis, Validation, Investigation, Writing - original draft, Writing - review and editing; Elisabete Augusto, Data curation, Formal analysis, Investigation, Methodology, Writing - original draft, Writing - review and editing; Vladimir Kouskoff, Formal analysis, Investigation, Methodology; Tiago Campelo, Investigation, Visualization, Methodology, Writing - review and editing; Christelle Martin, Resources, Writing - original draft, Project administration; Yann Humeau, Resources, Supervision, Methodology, Writing - original draft; Nicolas Chenouard, Software, Investigation, Methodology, Writing - original draft; Frederic Gambino, Conceptualization, Resources, Software, Formal analysis, Supervision, Funding acquisition, Validation, Investigation, Methodology, Writing - original draft, Project administration, Writing - review and editing

### Author ORCIDs
Frederic Gambino (iD) https://orcid.org/0000-0002-2981-5030

### Ethics
Animal experimentation: All experiments were performed in accordance with the Guide for the Care and Use of Laboratory Animals (National Research Council Committee (2011): Guide for the Care and Use of Laboratory Animals, 8th ed. Washington, DC: The National Academic Press.) and the European Communities Council Directive of September 22th 2010 (2010/63/EU, 74). Experimental protocols were approved by the institutional ethical committee guidelines for animal research (N° 50DIR_15-A) and by the French Ministry of Research (N°02169.01).

### Decision letter and Author response
Decision letter https://doi.org/10.7554/eLife.62594.sa1
Author response https://doi.org/10.7554/eLife.62594.sa2

## Additional files

### Supplementary files
• Transparent reporting form

### Data availability
All data generated or analysed during this study are included in the manuscript and supporting files. All numerical data are provided in the source tables.

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
