## [Decision Letter]

**Acceptance summary:**

The manuscript presents a large set of technically challenging and impressive experiments addressing the cellular and circuit mechanisms of discriminative fear learning in the frontal association cortex. The results are likely to be of interest to those studying fear conditioning, auditory learning, corticolimbic dynamics, and related psychiatric conditions.

**Decision letter after peer review:**

Thank you for submitting your article "The integration of Gaussian noise by long-range amygdala inputs in frontal circuit promotes fear learning" for consideration by *eLife*. Your article has been reviewed by three peer reviewers, and the evaluation has been overseen by a Reviewing Editor and Kate Wassum as the Senior Editor. The reviewers have opted to remain anonymous.

The reviewers have discussed the reviews with one another and the Reviewing Editor has drafted this decision to help you prepare a revised submission.

Summary:

The manuscript by Aime et al. presents a large sets of technically challenging and impressive experiments studying the cellular and circuit mechanisms of discriminative fear learning in the frontal association cortex (FrA). Using recording and modulatory approaches, the authors show that white Gaussian noise (WGN) causes greater depolarization than pure tones, more local dendritic events than pure frequency tones in both the anaesthetized and awake preparation, is differentially modulated by photo-inhibition of FrA during discriminative conditioning (when in the role of CS-). Further, axons within the FrA from the basolateral amygdala (BLA) increased activity between conditioning trials, and predicted the level of auditory fear learning. The topic is of broad interest and the approach taken considers both the cellular and circuit dynamics underlying discriminative fear learning. The authors have addressed the concerns of the reviewers in a previous submission of the manuscript, however, some outstanding concerns remain (detailed below) which we believe can be addressed through revisions in the text.

Essential revisions:

1) One of the main concerns echoed by all reviewers (and was also something noted in the previous version of the manuscript) was the idea of "unique properties of WGN over frequency modulated and pure tone frequency". This appears to be true for the stimuli tested, however, the chirps and pure frequencies did not cover the same frequency range as the WGN. Put simply, the WGN included frequencies up to 20 kHz, whereas the pure tones and linear chirps only involved frequencies between 4 and 8 kHz. Also, although the chirp stimuli was more complex (and probably involves more cortical processing than pure tones), it is still not comparable to WGN (as clearly shown in Figure 1—figure supplement 1). Therefore, it remains that the 'unique properties' of the WGN stimuli that result in greater neural activity might simply be due to the different frequencies and complexity of the auditory stimuli used. This needs to be made explicit in the manuscript.

2) In Figure 1, FrA neurons decrease voltage under NMDAR blockade. The authors state that “this suggests that a fraction of the NMDAR conductances recruited by auditory stimulations is masked by non-specific inhibition under control conditions”. However, the “hyperpolarization” is reported from the voltage change during NMDA block (i.e. zeroed to baseline). Couldn't the inhibition reported also be a reflection of the decrease in conductance through NMDA channels (and other Ca^2+^ dependent channels)?

3) The finding that the activation of BLA-to-FrA axons supports the non-linear integration of auditory tones is interesting. However, non-linear integration typically leads to the generation of action potentials. Clarification on whether there was a change in evoked action potentials during the combined stimulus is necessary. In addition, please clarify why when paired with activation of BLA axons, both WGN and pure tones resulted in a larger than expected change in membrane voltage given that pure tones are not encoded in the PFC.

4) The authors conclude that "our data indicate that BLA-to-FrA synapses are likely to be weak and unreliable". This conclusion is surprising based on the evoked responses recorded in vitro (Figure 5—figure supplement 1), especially considering the input presumably lands on the distal dendrites of layer 2/3 pyramidal neurons. Was the response modulated by different intensities of the photo-activating LED light?

5) Why does the influence of the auditory stimulus significantly outlast the stimulus presentation? Does the difference between CS- and CS+ in the learner mice return to “unlearnt” (naïve) mice over time?

6) The behavioural/optogenetic results clearly show the influence of WGN is specific to CS-. However, the BLA-FrA axonal imaging data is less convincing. Could BLA-FrA axons be signaling the contextual threat of the situation with no specificity to CS+ or CS-?

7) WGN dendritic and somatic activity is reduced after fear conditioning. The authors suggest this is due to plasticity however since the encoding of WGN is similar to pure tones after conditioning. Wouldn't this make more difficult to discriminate?

8) Figure clarification. It is difficult to compare the Ca^2+^ activity based on events / dendrite in Figure 2. Please convert to frequency or a measure that can be compared with other conditions. Aside from the fwhm, was there any difference in the waveform (amplitude/duration) of the evoked Ca^2+^ between the different auditory stimuli? Overall, Figure 3 is difficult to interpret. 3D and 3H should be split into different graphs for protocol type, as it is currently impossible to separate the conditions. In Figure 4F, when was the stimulus presented? Why is the cumulative distribution taken over 40 seconds? How was the cumulative ∆F/F calculated?

9) Are the fear conditioning protocols equally effective in inducing fear memories? Compared to control mice (Figure 3), there appears to be a big difference in the GFP controls during the different protocols in Figure 6. Was this significant?

---

## [Author Response]

Essential revisions:1) One of the main concerns echoed by all reviewers (and was also something noted in the previous version of the manuscript) was the idea of "unique properties of WGN over frequency modulated and pure tone frequency". This appears to be true for the stimuli tested, however, the chirps and pure frequencies did not cover the same frequency range as the WGN. Put simply, the WGN included frequencies up to 20 kHz, whereas the pure tones and linear chirps only involved frequencies between 4 and 8 kHz. Also, although the chirp stimuli was more complex (and probably involves more cortical processing than pure tones), it is still not comparable to WGN (as clearly shown in Figure 1—figure supplement 1). Therefore, it remains that the “unique properties” of the WGN stimuli that result in greater neural activity might simply be due to the different frequencies and complexity of the auditory stimuli used. This needs to be made explicit in the manuscript.

We agree with the reviewers that the dendritic mechanism we describe is very likely not restricted to WGN during learning and might apply to other complex tones including natural tones. Thus, to make it more explicit we have discarded the term “unique”, and we have added the following sentences in the first and second paragraphs of the Discussion:

“Thus, despite the dendritic and somatic properties of WGN that we observed in the FrA (Figure 1—figure supplement 1), it remains likely that other complex tones, including natural tones or more complex frequency-modulated tones covering a wider spectrum of pure frequencies, could also facilitate learning in a similar way than WGN”.

“… In support of our hypothesis, we found that simple linear chirps produced more somatic depolarization than pure tones. This further suggests that more complex frequency-modulated tones could theoretically also activate FrA pyramidal neurons and affect fear learning like WGN did”.

2) In Figure 1, FrA neurons decrease voltage under NMDAR blockade. The authors state that “this suggests that a fraction of the NMDAR conductances recruited by auditory stimulations is masked by non-specific inhibition under control conditions”. However, the “hyperpolarization” is reported from the voltage change during NMDA block (i.e. zeroed to baseline). Couldn't the inhibition reported also be a reflection of the decrease in conductance through NMDA channels (and other Ca^2+^ dependent channels)?

We have performed experiments during which we presented different tones on the same cell before and after the blockade of NMDAR (Figure 1—figure supplement 2G-I). This figure shows single cell example of membrane fluctuation upon WGN (G, left) and 8 kHz (G, right) before and after dAP5 application (colored bars represent averaged Vm over 1 sec time window).

Tones’ presentation hyperpolarized membrane potential when NMDAR are blocked (G-H, middle), whilst 10 sec before auditory stimulation Vm is not affected while NMDAR are already blocked. We concluded that the hyperpolarization depended on effects (such as local inhibitory balance) that are counteracted by NMDAR conductances under baseline conditions. During NMDAR block, this “inhibitory” component becomes visible, and probably reflect a classic feedforward circuit.

However, we have not tested whether this effect was effectively mediated by inhibition, and we agree with the reviewers that alternative explanations still exist. Therefore, we have modified the sentence: “This suggests that a fraction of the NMDAR conductances recruited by auditory stimulations is masked under control conditions, and that both WGN and pure tones might activate dendrites”.

3) The finding that the activation of BLA-to-FrA axons supports the non-linear integration of auditory tones is interesting. However, non-linear integration typically leads to the generation of action potentials. Clarification on whether there was a change in evoked action potentials during the combined stimulus is necessary.

The BLA+WGN nonlinearities were much stronger than those generated by BLA+8 kHz. However, it failed to produce any somatic action potential in FrA pyramidal neurons during the stimulation. This is now clearly stated in the Results and Discussion.

We agree that the supra-linear summation of convergent excitatory inputs facilitate the propagation of distal dendritic spikes, which might eventually support the generation of somatic action potential^1,2^. Nevertheless, while the absence of spiking could have been caused by the anesthesia, dendritic spikes and non-linear integration also support other functions, and might induce for example long-lasting changes in synaptic strength and intrinsic excitability^1,3–7^. As an example, we have shown that the non-linear integration of distinct thalamic inputs does not lead to somatic action-potential in cortical pyramidal, but facilitates instead synaptic plasticity^3^.

We now acknowledge the fact that the function of these BLA-mediated nonlinearities remains unclear during learning (“The mechanism by which these BLA-mediated nonlinearities affect learning remains unclear”). We also have modified the third paragraph of our Discussion to introduce these two alternatives:

“…it is possible that the modest activation of BLA synapses in FrA apical dendrites facilitates or gates the propagation towards the soma of tone-evoked dendritic events which could enhance the generation of somatic action potential ^1,2^. […] Nevertheless, although the BLA+WGN nonlinearities were much stronger than those generated by BLA+8 kHz, it failed to produce any somatic action potential in FrA pyramidal neurons”.

“While the absence of spiking could have been caused by the anesthesia, another explanation would be that the non-linear interaction between compartmentalized streams of neural activity induces long-lasting changes in synaptic strength and intrinsic excitability ^1,3–7^. […] Although this effect was analyzed no longer than 30 sec after the end of the stimulation, it also prompts the speculation that the lasting alteration of FrA membrane potential evoked by the BLA+*WGN* non-linearities might reflect a global brain state change which would modify the integration of future incoming sensory inputs, such as it occurs during consecutive pairing upon conditioning”.

In addition, please clarify why when paired with activation of BLA axons, both WGN and pure tones resulted in a larger than expected change in membrane voltage given that pure tones are not encoded in the PFC.

Our data indicate that pure tones hyperpolarized pyramidal neurons in the presence of dAP5 or iMK801, thus revealing a small NMDAR component that was masked under control conditions (please see point 2 above). We have added the following sentence in the Results section: “This suggests that a fraction of the NMDAR conductances recruited by auditory stimulations is masked under control conditions, and that both WGN and pure tones might activate dendrites. However, on average, the NMDAR-mediated component of the evoked cVm change was much larger in response to WGN…”.

We believe that this small activation during pure tone presentation can be potentiated when BLA inputs are co-activated, although to a much lesser extent as compared to WGN. Altogether, our data indicate that both input streams together are required to generate non-linearity, and that “BLA axons are necessary to produce non-linear integration of auditory inputs in FrA neurons, which is stronger for Gaussian noise than for pure tones”.

4) The authors conclude that "our data indicate that BLA-to-FrA synapses are likely to be weak and unreliable". This conclusion is surprising based on the evoked responses recorded in vitro (Figure 5—figure supplement 1), especially considering the input presumably lands on the distal dendrites of layer 2/3 pyramidal neurons. Was the response modulated by different intensities of the photo-activating LED light?

This is an interesting point. Several possibilities could explain the difference between in vivo and in vitro experiments. The first perhaps lies in the nature of the stimulation (LED have relatively high divergence, causing the power of the LED to be spread across the field of view). Second, we have used in vivo the same laser intensity than for behavioral experiments. Although this was not necessarily the most effective intensity, we haven’t tested higher intensities for the sake of consistency between experiments. Third, and most importantly, it is known that cortical neurons in vivo are spontaneously active both under anesthesia and during quiet wakefulness, with specific pattern of up and down spontaneous states that might profoundly affect the responses to specific stimulations^8^. For example, in the barrel cortex, spontaneous up states inhibit sensory-evoked responses, which are smaller, briefer and spatially more confined than during down states^9^. In agreement, we observed that “BLA-to-FrA inputs were mostly undetectable in vivo and became visible only when the stimulation was delivered during down states, suggesting that BLA-to-FrA inputs in vivo are likely to be modified by on-going spontaneous activity”.

We also have clarified these points in the second paragraph of the Discussion: “However, and in contrast to what we observed in vitro, the photo-stimulation of ChR2-expressing BLA neurons in vivo at an intensity similar to the one used in behavioral experiments, produced depolarizations in FrA pyramidal neurons that were rather weak and unreliable. The difference between in vitro and in vivo conditions may reflect the interaction between evoked responses and on-going spontaneous activity that occurs in vivo^8^”.

5) Why does the influence of the auditory stimulus significantly outlast the stimulus presentation? Does the difference between CS- and CS+ in the learner mice return to “unlearnt” (naïve) mice over time?

This is also an interesting point. While the mechanisms remain unclear, it has been shown that cortical circuits with local recurrent connections can generate self-sustaining activity that can be turned on for long period of time by brief synaptic inputs, causing possibly changes in the global brain state (such as it occurs during switching from asleep to conscious)^8,10–12^. For example, in vivo recordings of striatal neurons in urethane-anaesthetized rats revealed that brief non-physiological stimuli cause a long-lasting depolarized state with fast, low-amplitude modulations that persisted until the cortex resumed ~1 Hz synchronous activity^11^. We have added the following sentence in the second paragraph of the Discussion to introduce this possibility: “These depolarizations persisted beyond the end of the stimulation, possibly reflecting changes in the dynamics of network activity through local recurrent connections”.

The second question, while interesting, is difficult to address with our current data set. We did not record somatic depolarizations from mice several days after learning, and we cannot answer whether the difference between CS- and CS+ in mice that learned (Figure 3—figure supplement 3) returns to baseline level observed in naïve mice, in other words if fear extinction also affects FrA pyramidal neurons by opposite mechanisms. Nonetheless, a comment on the results description was added to leave this as an open question: “Whether the effect of learning on FrA pyramidal neurons is cancelled following fear extinction remains unknown and would be interesting to explore”.

6) The behavioural/optogenetic results clearly show the influence of WGN is specific to CS-. However, the BLA-FrA axonal imaging data is less convincing. Could BLA-FrA axons be signaling the contextual threat of the situation with no specificity to CS+ or CS-?

The encoding of context representation is supposed to involve hippocampal-BLA communication during learning^13^. Whether the BLA sends this information to the FrA remains unknown. However, we found that “the averaged cumulative ΔF/F0 measured during CS- was always higher than during CS+ …, revealing that the overall activity of BLA-to-FrA axons was stronger between conditioning trials, notably at the time when CS- occurred”.

We believe these data go against the signaling of the contextual threat as one would expect in that case no difference between CS+ and CS-. However, answering this question would require to image the same set of axons before/after learning in the same conditioning context. Given that we did not perform these specific experiments, this possibility remains and we have added the following sentence in the Discussion: “Although it remains unknown whether BLA axons might convey contextual information to the FrA, they seem to transmit integrated information about the association itself…”.

7) WGN dendritic and somatic activity is reduced after fear conditioning. The authors suggest this is due to plasticity however since the encoding of WGN is similar to pure tones after conditioning. Wouldn't this make more difficult to discriminate?

This is an important point. From our point of view, the fact that after learning, the dendritic activity triggered by WGN and pure tone is similar (as captured by dendritic calcium imaging in Figure 3—figure supplement 2 and somatic whole-cell recordings in Figure 3—figure supplement 3) does not mean that the local network has less capacity to discriminate both stimuli after learning, but instead that it has less capacity to learn again from the same stimuli after conditioning. In fact, cVm change after conditioning correlates with the learning index (Figure 3—figure supplement 3D). If the changes observed were correlated with poor discrimination between CS+ and CS-, one would expect a correlation with fear generalization or absence of learning, instead of learning. In that context, the non-linear integration of WGN and BLA inputs would not be critical after learning once memories are formed, but might facilitate during learning the recruitment of FrA neurons into memory traces.

Nevertheless, the question of how dendritic activity is translated during or shortly after learning into action potential (in other words, how pyramidal neurons are recruited into memory traces) is impossible to address with our current data set, and would require the chronic imaging of FrA activity at the population level. We have modified the manuscript in order to accommodate this point:

“In addition, WGN-induced somatic plateau potentials were reduced in conditioned animals in a learning-dependent manner, suggesting that NMDAR-dependent plasticity mechanisms in FrA neurons were engaged during learning (Figure 3—figure supplement 3). […] This suggest that, while dendritic signaling appears critical during learning to recruit pyramidal neurons into memory traces, other circuit and cellular mechanisms are at play to discriminate sensory cues after learning”.

“While this mechanism would occur only if WGN is presented between conditioning trials (and thus used as CS-), it might eventually facilitate the recruitment of neurons into specific cue memory traces”.

8) Figure clarification. It is difficult to compare the Ca^2+^ activity based on events / dendrite in Figure 2. Please convert to frequency or a measure that can be compared with other conditions.

We have modified Figure 2 accordingly.

Aside from the fwhm, was there any difference in the waveform (amplitude/duration) of the evoked Ca^2+^ between the different auditory stimuli?

The amplitudes of calcium transient were not different between the different tones. We now provide these data in the Figure 2.

Overall, Figure 3 is difficult to interpret. Figure 3D and H should be split into different graphs for protocol type, as it is currently impossible to separate the conditions.

We have modified Figure 3 to split experimental conditions into different panels. The manuscript and Figure 3—figure supplement 1 have been adjusted accordingly.

In Figure 4F, when was the stimulus presented? Why is the cumulative distribution taken over 40 seconds? How was the cumulative ∆F/F calculated?

We have modified Figure 4F to include the stimulus epoch (0-30 sec). The cumulative ∆F/F was calculated over 40 sec but the analysis was done at 30 sec (end of the stimulation).The cumulative ∆F/F corresponds to the integral of all calcium transients detected during this period of time.

9) Are the fear conditioning protocols equally effective in inducing fear memories? Compared to control mice (Figure 3), there appears to be a big difference in the GFP controls during the different protocols in Figure 6. Was this significant?

This is an important point. To exclude any human or experimental bias, we have performed and analyzed the optogenetic experiments blind to the genotype (GFP or ArchT). Protocols 1 and 2 are indeed not equally effective. We pooled all our control mice together (Figure 6—figure supplement 1D-E) and freezing responses after protocol 2 were significantly lower than after protocol 1, but were similar to those produced after protocol 1 when CS- was omitted during learning.

This suggests that: (1) fear conditioning with protocol 1 (CS+: 8kHz tone, CS-: WGN) tends to produce more fear responses during recall than with protocol 2; (2) WGN, when used as CS- during conditioning, facilitated learning. We have extended the manuscript on these points (Discussion, Materials and methods).

References:

1) Jarsky, T., Roxin, A., Kath, W. L. and Spruston, N. Conditional dendritic spike propagation following distal synaptic activation of hippocampal CA1 pyramidal neurons. Nat. Neurosci. 8, 1667–1676 (2005).2) Palmer, L. M. et al. NMDA spikes enhance action potential generation during sensory input. (2014) doi:10.1038/nn.3646.3) Gambino, F. et al. Sensory-evoked LTP driven by dendritic plateau potentials in vivo. Nature 515, (2014).4) Larkum, M. A cellular mechanism for cortical associations: an organizing principle for the cerebral cortex. Trends Neurosci. 36, 141–51 (2013).5) McGaugh, J. L. Making lasting memories: remembering the significant. Proc. Natl. Acad. Sci. U. S. A. 110 Suppl 2, 10402–7 (2013).6) Dudman, J. T., Tsay, D. and Siegelbaum, S. A. A role for synaptic inputs at distal dendrites: instructive signals for hippocampal long-term plasticity. Neuron 56, 866–79 (2007).7) Xu, N. et al. Nonlinear dendritic integration of sensory and motor input during an active sensing task. Nature 492, 247–51 (2012).8) Ferezou, I. and Deneux, T. Review: How do spontaneous and sensory-evoked activities interact? Neurophotonics 4, 031221 (2017).9) Petersen, C. C. H., Hahn, T. T. G., Mehta, M., Grinvald, A. and Sakmann, B. Interaction of sensory responses with spontaneous depolarization in layer 2/3 barrel cortex. Proc. Natl. Acad. Sci. 100, 13638–13643 (2003).10) Shu, Y., Hasenstaub, A. and McCormick, D. A. Turning on and off recurrent balanced cortical activity. Nature 423, 288–293 (2003).11) Kasanetz, F., Riquelme, L. A. and Murer, M. G. Disruption of the two-state membrane potential of striatal neurones during cortical desynchronisation in anaesthetised rats. J. Physiol. 543, 577–589 (2002).12) LaBerge, D. Sustained attention and apical dendrite activity in recurrent circuits. Brain Research Reviews vol. 50 86–99 (2005).13) Maren, S., Phan, K. L. and Liberzon, I. The contextual brain: Implications for fear conditioning, extinction and psychopathology. Nature Reviews Neuroscience vol. 14 417–428 (2013).